# CODE WORLD MODELS FOR GENERAL GAME PLAYING

**Wolfgang Lehrach**[*]   **Daniel Hennes**[*]   **Miguel Lázaro-Gredilla**[*]
**Xinghua Lou   Carter Wendelken   Zun Li   Antoine Dedieu**
**Marc Lanctot   Atil Iscen   John Schultz   Marcus Chiam**
**Ian Gemp   Piotr Zielinski   Satinder Singh   Kevin P. Murphy**
Google DeepMind
{wpl, hennes, lazarogredilla, xinghua, cwendelken, lizun, adedieu,
lanctot, atil, jhtschultz, marcuschiam, imgemp, zielinski, baveja, kpmurphy}@google.com

## ABSTRACT

Large Language Models (LLMs) reasoning abilities are increasingly being applied to classical board and card games, but the dominant approach—involving prompting for direct move generation—has significant drawbacks. It relies on the model's implicit fragile pattern-matching capabilities, leading to frequent illegal moves and strategically shallow play. Here we introduce an alternative approach: We use the LLM to translate natural language rules and game trajectories into a formal, executable world model represented as Python code. This generated model—comprising functions for state transition, legal move enumeration, and termination checks—serves as a verifiable simulation engine for high-performance planning algorithms like Monte Carlo tree search (MCTS). In addition, we prompt the LLM to generate heuristic value functions (to make MCTS more efficient), and inference functions (to estimate hidden states in imperfect information games). Our method offers three distinct advantages compared to directly using the LLM as a policy: (1) Verifiability: The generated CWM serves as a formal specification of the game's rules, allowing planners to algorithmically enumerate valid actions and avoid illegal moves, contingent on the correctness of the synthesized model; (2) Strategic Depth: We combine LLM semantic understanding with the deep search power of classical planners; and (3) Generalization: We direct the LLM to focus on the meta-task of data-to-code translation, enabling it to adapt to new games more easily. We evaluate our agent on 10 different games, of which 4 are novel and created for this paper. 5 of the games are fully observed (perfect information), and 5 are partially observed (imperfect information). We find that our method outperforms or matches Gemini 2.5 Pro in 9 out of the 10 considered games.

## 1 INTRODUCTION

Large Language Models (LLMs) have shown impressive abilities at solving various reasoning tasks, and recently have been applied as "agents" which can play classical (often multi-player) games, like Chess, Go, and even complex imperfect information games like Poker and Bridge. The standard approach is to treat the LLM as a policy, by asking it to pick a move at each step using a prompting strategy based on the trajectory of observations and actions seen so far, plus optional text meta data about the game. This method treats the LLM as an end-to-end "intuitive player", leveraging its vast training data to recognize patterns and select moves that seem promising. However, strategic mastery often requires deep multi-step lookahead, characteristic of a "System 2" deliberation (Kahneman, 2003). While strong play can be achieved through training *specialist* models (Ruoss et al., 2024; Schultz et al., 2025), direct play from generalist LLMs often lacks deep tactical foresight, despite recent advances in "thinking" (Liao et al., 2025), as we show empirically in this paper. In addition,

---

[*]Equal contribution.
Note that Jordi Grau-Moya (jordigrau@google.com) is the 8th author but due to a purely administrative oversight was left off ICLR submission. Jord is present on the contemporary arxiv submission.

the LLM as policy approach does not work very well on novel games that are not part of the LLM's training set, as we will also show.

We propose to use LLMs in a different way, namely as induction engines that can leverage their prior knowledge to map a small amount of observed trajectory (game play) data, plus a textual game description, into plausible world models, represented as Python code, using iterative code refinement methods as in Tang et al. (2024a). We call the result of this process a "Code World Model" (CWM). In the context of game playing, a CWM consists of a definition of the (possibly latent) state, a function that specifies which moves are legal at each step, a state transition function, an observation function (for latent states), a reward function, and a function that checks for termination. Furthermore, for the challenging case of partially observable games, we introduce a novel paradigm that effectively tasks the LLM with synthesizing a regularized autoencoder: an inference function (the encoder) maps observations to plausible latent histories, and the CWM (the decoder) reconstructs observations from them, with the game's rules and API serving as a strong structural regularizer.

Although there is prior work that uses LLMs to learn symbolic world models (see Sec. 3), and then leverage them for planning, we differ in three main ways. First, we handle the case of partially observed and stochastic worlds (such as Poker), whereas all prior work (to the best of our knowledge) either assumes fully observed and deterministic environments, or (in the case of Curtis et al. (2025)) assumes post-hoc observability; both cases make model learning much easier. Second, in addition to learning a CWM, we ask the LLM to generate heuristic value functions, which significantly improves the performance of our search-based policies, such as MCTS and Information Set MCTS (Cowling et al., 2012). Third, we demonstrate that our approach outperforms a state-of-the-art "thinking" LLM across various two-player games, including novel (or "OOD") ones which we create, to avoid contamination issues with the training set of the LLM.

## 2 BACKGROUND

Interactions in multiplayer games can be described using the formalism of extensive-form games (Kuhn, 1953; Shoham & Leyton-Brown, 2009; Albrecht et al., 2024; Murphy, 2025): there is a set $\mathcal{N} = \{1, 2, \cdots, n\}$ of $n$ players that take discrete actions $a \in \mathcal{A}$. Sequences of actions are called histories $h \in \mathcal{H}$; all games start at the initial empty history, and end at terminal histories $\mathcal{Z} \subseteq \mathcal{H}$. There is a special player called *chance* (also sometimes called nature), $c$, which plays with a known, fixed (stochastic) policy—the chance outcome distribution—*e.g.*, representing dice rolls and card draws. Due to chance events being explicitly represented by the game environment, each history $h$ can be thought of as a unique transcription of a game (either finished or in progress) and as a "ground truth" state known only to the environment. At every history $h$, there is a player to act $\tau(h) \in \mathcal{N} \cup \{c\}$, and a set of legal actions $\mathcal{A}(h) \subseteq \mathcal{A}$. Formally defining states in partially-observable (imperfect information) requires several definitions; thus, we defer this to Appendix C to couple it with the description of the search method (policy generation). Agents encode policies to take actions $\pi(h) \in \Delta(\mathcal{A})$, where $\Delta(\cdot)$ represents a discrete probability distribution. For each agent $i$, the goal is to find a policy that maximize its own cumulative reward $\sum_{t=1}^{T} r^i(h_t)$. However, in the multiagent setting each individual objective jointly depends on choices of other agents.

Our game environments are based on OpenSpiel (Lanctot et al., 2019): each implementation provides logic to determine legal actions, transitions from one ground truth state to the next, rewards, and player observations in a general way. However, the agent does not know the true environment model. Instead, it must learn the code world model by using an LLM applied to a text description of the game, together with example game play data, as described in detail in Sec. 4. Given the learned CWM, we pick the best move by using existing game solvers: for perfect information games, we use MCTS, and for imperfect information games, we use Information Set MCTS (see Appendix C). In both cases, we optionally augment the search algorithm with a learned value function, and in the case of ISMCTS, we augment the search algorithm with a hidden state estimator. We also tried learning a policy using PPO applied to the (partially observed) CWM: see Appendix E for details.

## 3 RELATED WORK

There is a growing interest in evaluating the abilities of LLMs to play games, as exemplified by the recent release of Kaggle Game Arena[1], as well as other recent work (Costarelli et al., 2024; Duan et al., 2024; Verma et al., 2025; Hu et al., 2025a; Sun et al., 2025; Cipolina-Kun et al., 2025; Hu et al., 2025b; Guertler et al., 2025; Mishra et al., 2025). Similar to these papers, our aim is to design LLM-based agents that play text-based games. Furthermore, like ggbench (Verma et al., 2025), we assess the generality of our agents using novel games, that are (by construction) out-of-distribution (OOD) for the LLM. However, rather than using the LLM directly as a policy, we focus on using the LLM to generate a CWM, to which we then apply standard solvers, such as (IS)MCTS or PPO.

There are a few other papers that also use a model-based approach, similar to ours. "WorldCoder" generates a set of CWM hypotheses from trajectory data using LLM-powered code synthesis, stores each hypothesis (candidate model) in a tree, and uses Thompson sampling to decide which hypothesis to ask the LLM to improve, see (Tang et al., 2024a). Given the learned CWM, WorldCoder uses ReAct-style methods (Yao et al., 2022) for decision-making. GIF-MCTS (Dainese et al., 2024) developed a similar method, but uses MCTS for agent decision-making. Our work extends this past work by considering strategic multiagent environments, synthesizing value functions (to speed up (IS)MCTS), and synthesizing and refining inference functions (to handle imperfect information games).

Imperfect information games can be considered a special kind of (multi-agent) partially observable Markov decision process (POMDP). Learning such models from observational data is notoriously difficult. In very recent work, Curtis et al. (2025) introduce "POMDP Coder", which learns a partially observed CWM. However, unlike us, they assume the hidden states are observed in hindsight (at the end of the trajectory). By contrast, we also consider a "closed deck" scenario, in which the hidden states are never observed. In addition, Curtis et al. (2025) use a determinized belief space planner (related to the POMCP method of Silver & Veness (2010)), whereas we use ISMCTS (see Appendix C) or PPO (Appendix E).

There are other many other ways to use LLMs for reasoning in games and multiagent systems. A recent line of work focuses on using LLMs to construct game-theoretic models of arbitrary scenarios in order to derive and deploy intelligent, strategic policies. Gemp et al. (2024) treats an LLM as an environment transition operator, controllable via instruction sets. An extensive-form game tree is explicitly constructed in OpenSpiel and an equilibrium over instruction sets is computed. Daskalakis et al. (2024) demonstrates how to design a game tree for Romeo and Juliet with the assistance of an LLM, subsequently modifying the tree so that the classic story lies in the support of its Nash equilibrium. Xu et al. (2025) embeds several observed Werewolf dialogues in a latent space, clusters the messages to form a finite action space and resulting game tree, and then runs counterfactual regret minimization on this discrete latent representation to derive a policy. Mensfelt et al. (2024a) proposed an approach to automatically translate natural language descriptions of small bimatrix games to logic representations (similarly in Mensfelt et al. (2024b)). Most closely related to this work, Deng et al. (2025) automated the construction of explicit (imperfect-information) extensive-form game trees from natural language descriptions of games, including a debugging module to ensure the resulting Gambit (Savani & Turocy, 2024) representation was valid. In contrast to this work, they only conditioned on game descriptions (rules) not observed trajectories and applied their pipelines to games with game trees containing at most 25 decision nodes (Kuhn Poker); code-world models offer the potential to scale to much larger game instances in some cases due to their more efficient encoding of repeat transitions.

## 4 METHODS

At a high level, when confronted with a new game, our general game playing agent follows these steps: First, it plays a few games to completion using a random policy. The data collected during each game forms a *trajectory*, which consists of observations, rewards, legal actions, and states at each timestep. Second, it uses a textual description of the rules of the game, plus the generated

---

[1] See https://www.kaggle.com/game-arena.

trajectories, to learn a CWM[2]. Finally, the agent plays the game in an arena against other opponents, using an MCTS policy built on top of the synthetic CWM. For imperfect information games (IIGs) we use ISMCTS instead of MCTS. If all the synthesized elements are correct, as the amount of play-time compute increases, the playing behavior of our agent gets closer to optimal. Thus, in contrast with LLM-as-a-policy agents, we shift the burden on the LLM from producing a good policy to producing a good world model, which in turn enables planning methods to turn compute into playing performance.

## 4.1 Synthesizing the Code World Model

A CWM is a playable, approximate copy of a target game. It contains functions providing logic to update the game state when an action is taken (transition function, which includes a termination), the legal actions given a state, the observation given a state (observations and state differ in the case of IIGs), the distribution for chance nodes, and the reward function for a state. All these functions are deterministic, with randomness entering the game only through the actions of the chance player. To synthesize a new CWM, we provide the LLM with the game's rules and offline trajectories, and demand that it creates a CWM following the OpenSpiel API (Lanctot et al., 2019) format. See Appendix H for prompt details.

A single-shot generation of the CWM will often be insufficient to produce a correct implementation of the game unless we add some kind of corrective feedback. Thus we subject the initial CWM to iterative *refinement* (Dainese et al., 2024; Tang et al., 2024b) to improve its quality. For refinement, a series of unit tests are automatically generated from the offline trajectories. For each transition in an offline trajectory, unit tests are generated in order to check the correctness of the CWM predictions as compared with the original trajectory (states, observations, rewards, legality of actions), and the absence of execution errors.

In the case of IIGs, this process requires that the offline trajectories contain not only the observations of the game and the actions of the players, but also the hidden states and the actions of all other players (including chance). The post-hoc availability of hidden states, an assumption also used in concurrent work (Curtis et al., 2025), can sometimes be unrealistic. Sec 4.4 introduces a novel approach to handle CWM learning from partially observed trajectories.

Unit tests are binary, so we can measure the *transition accuracy* as the rate of correctness of such tests. We refine the CWM until perfect transition accuracy (1.0) is achieved or our refinement budget runs out. We feed back the stack traces from failed unit tests to the LLM to help the refinement. We consider two separate approaches to refinement:

**Conversation (sequential refinement).** This is a serial "chat mode" approach, in which the stack trace of a newly failed unit test is appended to our previous interactions with the LLM to create the new prompt, and a new CWM addressing the unit test failure is requested. Failed unit tests derived from the offline trajectories are submitted to the LLM until all pass.

**Tree search.** Just like in the REx approach (Tang et al., 2024b;a), we maintain multiple CWMs in a refinement tree structure, and use Thompson sampling to choose which CWM to refine next, favoring those that either have high transition accuracy or have been refined few times. Each LLM call consists of a fresh prompt that contains the CWM chosen to be refined, the refinement instructions, and the stack trace of a failed unit test for that CWM. The prompts and hyperparameters used during synthesis are presented in Appendix H.

## 4.2 Synthesizing inference functions for IIGs

One of the novelties of our work is the synthesis of inference functions to enable the use of ISMCTS planning with the learned CWM at play time in imperfect information games (IIGs). To see why this is necessary, note that ISMCTS requires that at each game step $t$ the agent can estimate the hidden state of the game $s_t$, as explained in Appendix C. More precisely, at play time, agent $i$ must

---

[2]Note: We could potentially update the CWM after each step of game play, as we acquire new data, but in this paper, we learn the model up-front, given the initial offline trajectories and game description, for reasons of efficiency.

be able to sample from its belief state $p_M(s_t|o^i_{1:t}, a^i_{1:t})$, where $M$ is the estimated CWM[3]. Since exact inference incurs an exponential cost in the worst case, we ask the LLM to synthesize code to approximately sample from the posterior, utilizing only agent $i$'s actions $a^i_{1:t}$ and observations $o^i_{1:t}$ so far from the offline trajectory. We consider two alternative approaches to achieve this goal: hidden history inference and hidden state inference. We describe these below.

**Hidden history inference.** Since all the functions in the CWM are deterministic, the posterior over the hidden state $s_t$ can be obtained from the posterior over the action history $h_t$, which includes the actions of the chance player. In this approach, the agent controlling player $i$ asks the LLM to create a function that samples $\tilde{h}_t \sim p_M(h_t|o^i_{1:t}, a^i_{1:t})$. The CWM can then be used to execute $\tilde{h}_t$ and recreate a history of hidden states $\tilde{s}_{1:t}$ and observations $\tilde{o}^i_{1:t}$. A unit test is created for each time step $t$ in which player $i$ acts, verifying that the sampled values match the run time evidence (i.e., $\tilde{o}^i_t = o^i_t$ and $\tilde{a}^i_t = a^i_t$). This allows refinement (on the offline trajectories) to be applied to the inference function.

Once the refined inference function passes all unit tests[4] (i.e., *inference accuracy* is 1.0), we can claim that the sampled $\tilde{h}_t$ belongs to the support of $p_M(h_t|o^i_{1:t}, a^i_{1:t})$, and therefore, the $\tilde{s}_t$ generated by this process belongs to the support of $p_M(s_t|o^i_{1:t}, a^i_{1:t})$. Although this does not guarantee that $\tilde{s}_t$ is correctly distributed, the correct support is already very informative, given the extremely sparse support of state posteriors in games. Furthermore, this approach guarantees that the sampled posterior state $\tilde{s}_t$ is a valid CWM state. Note that *at play time* the (test) inference accuracy can drop below 1.0 (depending on how well the synthesized inference code generalizes to novel observations), meaning that the approximate posterior samples might not always belong to the support of the actual posterior. However, $\tilde{s}_t$ is still guaranteed by construction to be a valid hidden state in the CWM.

**Hidden state inference.** Rather than obtaining a state posterior sample indirectly through the action history, it is also possible to ask the LLM to create code that directly samples $\tilde{s}_t \sim p_M(s_t|o^i_{1:t}, a^i)$. Then, the CWM can be used to obtain $\tilde{o}_t$ from $\tilde{s}_t$. Correctness of the inference function can be partially validated by a unit test at each time step that verifies that the sampled values match the actual observations, $\tilde{o}_t = o_t$. CWM refinement can then be used to improve the synthesized inference function. State inference is potentially much simpler than full history inference, but it cannot guarantee that the produced sample $\tilde{s}_t$ belongs to the support of the posterior, nor that it constitutes a valid CWM hidden state, because it ignores the dependency between consecutive states.

## 4.3 Synthesizing value functions

Another novelty of our work is the synthesis of value functions to speed up and improve value estimation in MCTS and ISMCTS. This can be faster (and potentially more accurate) than estimating the value of a new leaf node through random rollouts. To synthesize a deterministic value function $V(s)$ to estimate the value of the (potentially hidden) state at leaf nodes, we can prompt the LLM to generate code, just as we did for learning the CWM. However, value functions are not refined, since there is no ground truth to compare to. Instead, multiple functions are generated and the best one is selected through a tournament.

## 4.4 Open deck vs closed deck during training

So far we assumed that the offline trajectories (used to train the CWM) contained hidden state information even for IIGs. Concurrent work Curtis et al. (2025) also assumes the ability to peek at hidden states. We refer to this setup as *open deck* synthesis[5]. This setup is justified in several practical scenarios, such as in a cooperative training environment where players share information to learn the mechanics of the game, during the design phase of a new game where developers have full

---

[3]For players other than $i$, we assume a uniform prior on the legal actions defined by the CWM. Only the support of this prior affects our approach, as we will focus on posterior support, see below.

[4]Unlike the CWM functions, inference functions are stochastic (samplers). Thus, their unit tests are potentially stochastic, but for correct inference functions they will deterministically pass.

[5]We want to emphasize that in our open deck setting, hidden state information is only available in the offline trajectories to aid CWM synthesis, and not during actual game play. Thus the players only ever see observations, but the CWM learner may see hidden states (in the open deck setting).

access to the state, or when a human expert provides fully annotated "open-book" demonstrations to bootstrap an agent's understanding.

However, there are scenarios in which the agent can only ever access its own observations and actions, so that the open deck assumption is violated. This would be the case, e.g., if the agent plays a novel game online. We refer to this scenario as *closed deck* synthesis; to the best of our knowledge, this scenario has not been addressed in prior CWM work.

To handle this scenario, we propose to combine the pieces introduced in the previous sections to build a regularized CWM "autoencoder". The idea is as follows: we ask the LLM to generate a CWM and a hidden history inference function, just like above, but we drop all the unit tests that are not verifiable without access to the hidden information (i.e., those checking the transition accuracy between consecutive hidden states), and we just keep the ones that we can verify (i.e., checking the result of mapping observations to hidden states and back to observations). We additionally add unit tests to a few iterations of random play ensuring that there are no execution errors. In other words, we refine based on the inference accuracy and lack of execution errors. This generates a kind of autoencoder, where the inference function acts as an encoder, producing a hidden sequence of actions $\tilde{h}_t$ from $o^i_{1:t}, a^i_{1:t}$ and the CWM acts as a decoder, recreating the observations and actions from the latent $\tilde{h}_t$. Instead of a bottleneck, or a regularization term, the game rules and the required OpenSpiel API (used in the unit tests) introduced in the context of the LLM act as regularizers to prevent trivial latent spaces from being discovered. Valid posterior histories $\tilde{h}_t$ (i.e., those that pass all unit tests) can be used to obtain a lower bound on the likelihood of the CWM, as follows: $p_M(o^i_{1:t}) = \sum_{h_t} p_M(o^i_{1:t}|h_t)p_M(h_t) \leq p_M(o^i_{1:t}|\tilde{h}_t)p_M(\tilde{h}_t) = p_M(\tilde{h}_t)$. (The last equality follows because $p_M(o^i_{1:T}|\tilde{h}_t) = 1$ when all unit test pass.) This lower bound is tightest when $\tilde{h}_t$ is the maximum a posteriori, but is valid for any sample.

## 5 EXPERIMENTS

Following the approach described in Sec. 4, we build an agent, which we call `CWM-(IS)MCTS`, which performs CWM synthesis (using either open or closed deck trajectories), and then plays using MCTS or ISMCTS. (We also tried learning a policy using PPO; see Appendix E for details.) We measure the playing abilities of our agent on multiple games against three other agents: A random legal action executor called `Random`; an (IS)MCTS agent that has access to the game's ground truth (GT) code, including inference functions but not value functions, which we call `GT-(IS)MCTS`; and an LLM as a policy, which we call `Gemini 2.5Pro` (we use "dynamic thinking", rather than specifying a thinking budget). All methods have access to the same data: the rules of the game as text and 5 offline trajectories. (IS)MCTS approaches always run 1,000 simulations before taking an action, using either the value function or 10 rollouts (in which all players act randomly) to determine the initial value of a new leaf node. A sketch of the information flow for each agent is given in Appendix G.

To validate the generality of our approach we use both perfect and imperfect information games, as well as well-known and OOD games. The perfect information games are: `Tic-tac-toe`, `Connect four`, `Backgammon`, `Generalized tic-tac-toe` (OOD), and `Generalized chess` (OOD). The imperfect information games are: `Leduc poker`, `Bargaining`, `Gin rummy`, `Quadranto` (OOD), and `Hand of war` (OOD). The out-of-distribution (OOD) games are not part of the LLM's training set, and have been created by us for these experiments. See Appendix I for the rules of each game.

### 5.1 SYNTHESIS ACCURACY

The CWM agent operates by synthesizing a CWM of the game (and potentially other auxiliary functions) prior to playing the game, see Sec. 4 for details. We use Gemini 2.5 Pro for synthesis. For the concrete prompts used during synthesis, see Appendix H. For examples of synthesized code, see Appendix J.

Refinement attempts to increase the fraction of units tests that pass, iterating until all pass or the budget for LLM calls is exhausted. The fraction of unit tests that the CWM passes is the *training transition accuracy*, and the fraction of tests that the inference function passes is the *training inference accuracy*. To check for overfitting to the offline trajectories, after synthesis, we measure the

accuracy on a separate test set of 10,000 transitions, randomly sampled from 100 games where each player is randomly assigned a random policy or MCTS on the ground truth game code. This yields the *test transition accuracy* and *test inference accuracy*. The test set is never used to train on; instead it is used to estimate the accuracy of the learned CWMs. Finally, at play time against the LLM as a policy, *online* transitions are observed, and again used to assess the accuracy of the CWM and inference functions.

### 5.1.1 PERFECT INFORMATION GAMES

For perfect information games, we find that we can learn a correct CWM for all the games, and that the resulting learned models have high test (generalization) accuracy. Both conversation and tree search work very well in this setting. Appendix D contains precise numbers (Tables 6 and 7, respectively), and shows the quick convergence of the CWM with the number of LLM calls (Fig. 6). We will stick with tree search for the remainder of this paper, since its ability to backtrack confers it additional resilience in harder settings.

### 5.1.2 IMPERFECT INFORMATION GAMES, OPEN DECK

In the case of imperfect information games (open deck learning), we find that the transition accuracy of the learned CWMs is very high, except for `Gin rummy`, where the training accuracy is just 84% and the test accuracy is 79%. See Table 1 for details. We hypothesize this is due to its high degree of logical and procedural complexity. Unlike games with more uniform rules, `Gin rummy` involves a multi-stage scoring phase (knocking, laying off melds, calculating deadwood, and checking for undercuts) that is difficult for the LLM to capture perfectly in code from a small number of trajectories. This highlights a key frontier for CWM synthesis: mastering games with intricate, multi-step procedural subroutines.

We also measure the inference accuracy obtained by the synthetic inference functions. We tried both hidden history and hidden state inference (see Sec.4.2). Results with hidden history inference (shown in Table 1 and Fig.1) are slightly better, so this will be the method of choice for the `CWM-ISMCTS` agent. (The results with hidden state inference are provided in Appendix D, Table 8 and Fig. 7.) Results for 3 of the 5 games are good, but once again we see that results for `Gin rummy` are quite poor (inference accuracy is only about 52%), and to a lesser extent `Hand of war` (inference accuracy is about 94%), even though CWM accuracy for `Hand of war` is good (about 98%). This suggests that hidden history inference is harder than learning the transition dynamics from a fully observed sequence of trajectories.

Table 1: Imperfect info. games, CWM refinement via tree search, hidden history inference.

| Game | OOD | transition accuracy | | | inference accuracy | | | # LLM calls |
|------|-----|------|------|--------|------|------|--------|------|
| | | train | test | online | train | test | online | |
| Bargaining | ✗ | 1.0000 | 0.9827 | 1.0000 | 1.0000 | 1.0000 | 1.0000 | 23.0 |
| Leduc poker | ✗ | 1.0000 | 0.9977 | 0.9942 | 1.0000 | 1.0000 | 1.0000 | 4.4 |
| Gin rummy | ✗ | 0.7816 | 0.7455 | 0.9044 | 0.5857 | 0.5376 | 0.9678 | 500.0 |
| Quadranto | ✔ | 1.0000 | 1.0000 | 1.0000 | 1.0000 | 0.9864 | 0.9916 | 6.0 |
| Hand of war | ✔ | 1.0000 | 0.9814 | 0.9868 | 1.0000 | 0.9357 | 1.0000 | 144.0 |

### 5.1.3 IMPERFECT INFORMATION GAMES, CLOSED DECK

Finally, we consider CWM synthesis with refinement in the novel closed deck setup in which no hidden information is available, not even post-hoc. The results in Table 2 show degradation on the synthesis quality with respect to the open deck setting of Table 1. Despite this, game play performance does not degrade significantly, as we show in the next section.

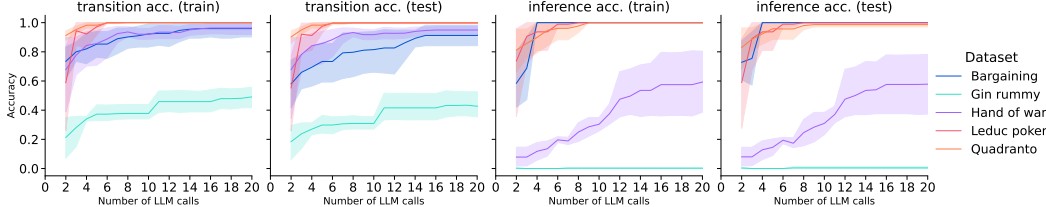

Figure 1: Evolution of the transition and inference accuracy with the number of LLM calls for imperfect games with refinement via tree search and hidden history inference.

Table 2: Imperfect information games, hidden history inference, closed deck.

| Game | OOD | inference accuracy | | | # LLM calls |
|------|-----|-------|------|--------|-------------|
| | | train | test | online | |
| Bargaining | ✗ | 1.00000 | 0.67359 | 0.76000 | 88.2 |
| Leduc poker | ✗ | 1.00000 | 0.97080 | 0.96585 | 9.0 |
| Gin rummy | ✗ | 0.05538 | 0.09523 | 0.53953 | 500.0 |
| Quadranto | ✔ | 1.00000 | 0.95183 | 0.96085 | 99.0 |
| Hand of war | ✔ | 0.86250 | 0.82130 | 0.94835 | 338.2 |

## 5.2 ARENA: GAME PLAY PERFORMANCE

In this section, we test how the previous synthesis results translate into playing performance against other opponents in our game arena. Since the CWM synthesis process is stochastic, we repeat it 5 times, automatically rejecting bad samples (see Appendix F), and pick a random CWM for each match. Results correspond to the average of 100 matches.

### 5.2.1 PERFECT INFORMATION GAMES

All of our perfect information games are ternary-outcome games, so we are limited to win, lose, or draw (W/L/D). Fig. 2 shows the performance of our `CWM-MCTS` agent, when acting as Player 0 or Player 1, against three different competitors. A player forfeits when it fails to provide a valid action in the allotted time. The middle pair of bars of each panel show `CWM-MCTS` playing against `GT-MCTS`, an upper bound for performance that uses the ground truth (GT) code of the game for planning. Both agents are similarly good, without either of them clearly winning in any of the games. This highlights the quality of our code synthesis. `CWM-MCTS` is able to beat `Gemini 2.5Pro` (which is used as a policy) in all the considered games. For detailed numerical results, see Table 9 in Appendix D. We used a synthetic value function for `Gen. tic-tac-toe`, see Fig. 9 for the ablation without value function.

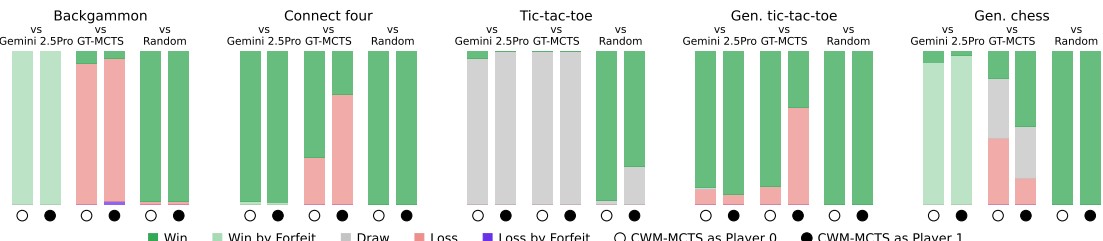

Figure 2: W/L/D rates for game play between `CWM-MCTS` and three opponents. CWMs are refined via tree search and hidden history inference.

### 5.2.2 IMPERFECT INFORMATION GAMES, OPEN DECK

Our imperfect information games contain a mixture of ternary-outcome games, zero-sum games and general-sum games (see Table 3 for a summary of all the games' characteristics). Win/loss/draw rates and payoff distributions are shown in Fig. 3. Except for `Hand of war`, `CWM-ISMCTS` beats or matches `Gemini 2.5Pro` in all imperfect information games. In the case of `Gin rummy`, this should be interpreted as `Gemini 2.5Pro` being a very weak player, you can check its forfeit rate in Table 16. For `Leduc poker`, although our average performance is superior, we also observe high variance. For `Bargaining` we used a synthetic value function, which results in a significant improvement when `CWM-ISMCTS` acts as player 1 (see Fig. 9 in Appendix D for the corresponding ablation). We did not observe an improvement or degradation in performance when value functions were applied to the other games.

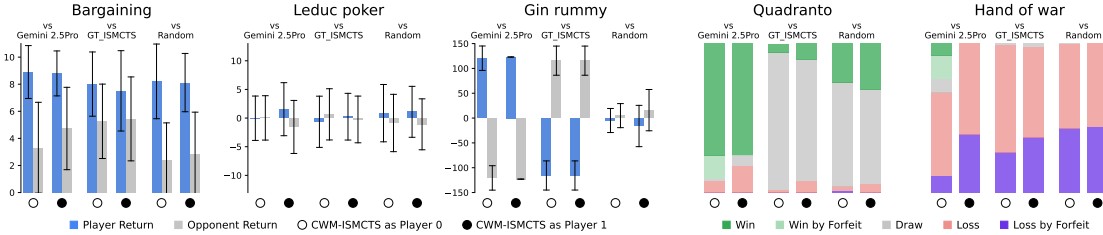

Figure 3: W/L/D rates and payoff distributions for game play between `CWM-ISMCTS` and three opponents. CWMs are refined via tree search and hidden history inference, *open deck*.

### 5.2.3 IMPERFECT INFORMATION GAMES, CLOSED DECK

Finally, we consider the closed deck setting, in which games are strictly partially observable, and no hidden state information or actions from other players are available in the offline trajectories. Results degrade w.r.t. the open deck setting, but `CWM-ISMCTS-Closed` continues to beat or match `Gemini 2.5Pro` (with high variance in the case of `Leduc poker`). We hypothesize that the non-intuitive improvement of `CWM-ISMCTS-Closed` at `Hand of war` w.r.t. the open deck setting could be due to the freedom to synthesize simpler state spaces when playing closed deck. Refer to Tables 14 and 15 in Appendix D for detailed results.

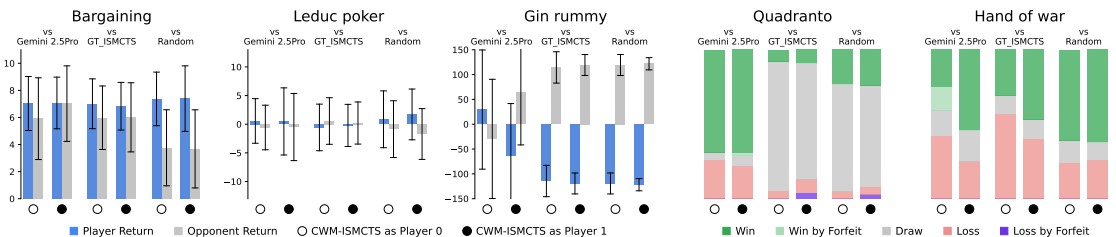

Figure 4: W/L/D rates and payoff distributions for game play between `CWM-ISMCTS` and three opponents. CWMs refined via tree search and hidden history inference, *closed deck*.

## 6 DISCUSSION

In this work we extend the existing CWM framework by considering two-player games, performing value function code synthesis to improve player performance, introducing the concept of "inference as code" to enable state estimation in imperfect information games, and providing a learning algorithm (based on code-based autoencoders) to enable learning in the novel closed deck (strict partial observability) setting. Our results show the superiority of this approach with respect to LLMs as policies on multiple perfect and imperfect information games, including newly created ones.

However, we also notice that our method struggles to learn the rules of `Gin rummy`, an imperfect information game with intricate logic, especially in the very challenging closed deck setting. In future work, we hope to extend our method to enable active and online learning of the world model, so the agent can more effectively discover the true hidden causal mechanisms underlying each game (c.f., (Geng et al., 2025)). In addition, we would like to extend the technique to handle open-world games with free-form text and/or visual interfaces, so as to evaluate it on larger sets of novel games, see (Ying et al., 2025).

ACKNOWLEDGMENTS

Jordi Grau-Moya is the 8th author but due to a purely administrative oversight was left off ICLR submission. Jordi is present in the contemporary arxiv submission and has been involved in this project since its inception, contributing significantly to both the foundational ideas and the codebase.

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

## A    INFORMATION ON THE GAMES

A summary of the games that we use in our experiments is given in Table 3.

Table 3: Details of the games that we use. The columns have the following meaning: OOD: whether the game is novel (no source code on the internet); Observability: Full means perfect information game, partial means imperfect information game; Payoff: W/L/D means Win/Lose/Draw, General means general sum; # Actions: number of possible actions; Obs. dim.: dimensionality of the observation tensor IS dim.: dimensionality of the information set (i.e., game's hidden state).

| Name | OOD | Observability | Payoff | # Actions | Obs. dim. | IS dim. |
|---|---|---|---|---|---|---|
| Backgammon | ✗ | Full | W/L/D | 1352 | 200 | |
| Connect four | ✗ | Full | W/L/D | 7 | 126 | |
| Tic-tac-toe | ✗ | Full | W/L/D | 9 | 27 | |
| Gen. tic-tac-toe | ✔ | Full | W/L/D | 36 | 108 | |
| Gen. chess | ✔ | Full | W/L/D | 5555 | 250 | |
| Bargaining | ✗ | Partial | General | 121 | 93 | 309 |
| Leduc poker | ✗ | Partial | Zero-sum | 3 | 16 | 30 |
| Gin rummy | ✗ | Partial | Zero-sum | 241 | 644 | 655 |
| Quadranto | ✔ | Partial | W/L/D | 5 | 9 | 7 |
| Hand of war | ✔ | Partial | W/L/D | 16 | 27 | 73 |

## B    SYNTHESIS RESULTS USING GEMMA27B

Table 4: Perfect information games, refinement via conversation using Gemma27B.

| Game | OOD | transition accuracy | | # LLM calls |
|---|---|---|---|---|
| | | train | test | |
| Backgammon | ✗ | 0.17462 | 0.16742 | 500.0 |
| Connect four | ✗ | 0.07826 | 0.10005 | 500.0 |
| Tic-tac-toe | ✗ | 0.98049 | 0.98441 | 193.0 |
| Gen. tic-tac-toe | ✔ | 0.94833 | 0.90307 | 500.0 |
| Gen. chess | ✔ | 0.49281 | 0.51603 | 500.0 |

Table 5: Imperfect information games, hidden history inference, closed deck using Gemma27B..

| Game | OOD | inference accuracy | | # LLM calls |
|---|---|---|---|---|
| | | train | test | |
| Bargaining | ✗ | 0.09412 | 0.14296 | 500.0 |
| Leduc poker | ✗ | 0.13333 | 0.11799 | 500.0 |
| Gin rummy | ✗ | 0.00797 | 0.01393 | 500.0 |
| Quadranto | ✔ | 0.09412 | 0.10343 | 500.0 |
| Hand of war | ✔ | 0.07812 | 0.07949 | 500.0 |

## C  Information Set Monte Carlo Tree Search

**Definition 1** (Extensive-Form Game with Imperfect Infomation (Osborne & Rubinstein, 1994)). *A finite **extensive-form game** with imperfect information is a tuple $(\mathcal{N}, \mathcal{A}, \mathcal{H}, \mathcal{Z}, \tau, \mathcal{I}, f_c, u_i)$, where:*

- *$\mathcal{N} = \{1, 2, 3, \cdots, n\} \cup \{c\}$ is a finite set of $n$ **players** and a special player called **chance**.*

- *$\mathcal{A}$ is a finite set of **actions**.*

- *$\mathcal{H}$ is a set of **histories** (sequences of actions), $\mathcal{Z} \subset \mathcal{H}$ the set of **terminal histories** (marking the full play of a game from start to finish). Every game starts at the empty history $h_0 = \emptyset$. At each non-terminal history $h \in \mathcal{H} - \mathcal{Z}$, let $A(h) \subseteq \mathcal{A}$ denote the set of legal actions available at $h$.*

- *A **player function** $\tau : \mathcal{H} - \mathcal{Z} \to \mathcal{N}$ that identifies which player is to act at every nonterminal history.*

- *For each player $i \in \mathcal{N}$, a partition $\mathcal{I}_i$ of $\{h \in \mathcal{H} - \mathcal{Z} : \tau(h) = i\}$ with the property that for all $I \in \mathcal{I}_i$, and $h, h' \in I$: $A(h) = A(h')$ and $\tau(h) = \tau(h') = i$. $\mathcal{I}_i$ is called player $i$'s **information partition** and each $I \in \mathcal{I}_i$ is called an **information state**.*

- *A function $f_c$ that assigns a probability distribution over actions at every $h \in \mathcal{H} - \mathcal{Z}$ where $\tau(h) = c$. Here, $f_c(h) \in \Delta(\mathcal{A})$ is the **chance outcome distribution** at chance event $h$.*

- *A **utility function** $u : \mathcal{N} \times \mathcal{Z} \to [U^-, U^+] \subset \Re$, where $U^-$ and $U^+$ are upper and lower bounds on the utility and $u_i(z)$ is the utility to player $i$ at terminal history $z$.*

Recall that a history encodes the sequence of actions taken by all players, *including chance*. But in an imperfect information game, not all aspects of the history are observable. For instance, in a game of poker, $h$ contains information about the cards held by all players (as chosen by the dealers actions), but some of this information is private and hence not known by some players. After an action is executed and added to the history $(h_{t-1}, a_t) \equiv h_t$, each player $i \in \mathcal{N}$ perceives individual observations $o_t^i(h_t)$. The state (from the perspective of an agent $i$) is then a function of $o_{1:t}^i$, e.g., just the last observation.

To choose actions in an IIG, we can use the Information Set MCTS method of (Cowling et al., 2012), which we now describe. First, recall that in classical MCTS, there is a root node corresponding to the current state of the game which all simulations start from, and non-root nodes which correspond to states that occur after the root state. At each node, statistics such as average values state-action values, $\hat{Q}(s, a)$, and simulation counts are maintained. The main differences in ISMCTS are: (i) the simulations start at a *distribution of possible ground truth states* and (ii) statistics are maintained and aggregated across information states with respect to the current player.

Figure 5 contains an example with a simplified poker game with a deck of three cards (Jack, Queen, King). In this example, the current player has received the King as a private card and no actions have yet been taken, so there are only two ground truth states: the opponent could have either the Queen or the Jack. An iteration first samples the Queen and continues with this ground truth state $h_0$, sampling actions, and generating histories $h_1, h_2, h_3$, and so on until the first node not in the tree is encountered. It is then added to the tree, and a random rollout policy takes over until a terminal state. The dotted boxes are the analogs of nodes stored in a tree (or lookup table) and correspond to information states. Return estimates (*i.e.*, Q-value statistics) and visit counts are maintained in these nodes as in classical MCTS (Coulom, 2007) (aggregated over different samplings of ground truth states), and UCB is used to select actions in the standard way (Kocsis & Szepesvári, 2006).

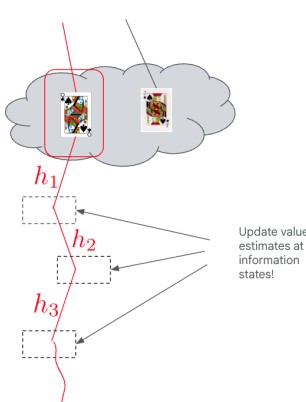

Figure 5: ISMCTS. A search tree is built over possible ground truth histories (e.g. $h_1$, $h_2$, …). Because the player cannot distinguish between certain histories, statistics are aggregated at the level of information sets (dotted boxes), which group all histories that appear identical to the player.

# D ADDITIONAL EXPERIMENTAL RESULTS

In the interest of space, some additional experimental results are included in this section.

## D.1 SYNTHESIS

### D.1.1 ACCURACY OF LEARNED TRANSITION AND INFERENCE FUNCTIONS

Comparing Table 6 and 7, it is apparent that even though both options work reasonably well, tree search has the edge, both in terms of accuracy (higher) and number of LLM calls (lower).

Table 6: Perfect information games, refinement via tree search.

| Game | OOD | transition accuracy | | | # LLM calls |
| --- | --- | --- | --- | --- | --- |
| | | train | test | online | |
| Backgammon | ✗ | 1.00000 | 0.99932 | 1.00000 | 16.8 |
| Connect four | ✗ | 1.00000 | 1.00000 | 1.00000 | 2.0 |
| Tic-tac-toe | ✗ | 1.00000 | 1.00000 | 1.00000 | 2.0 |
| Gen. tic-tac-toe | ✔ | 1.00000 | 1.00000 | 1.00000 | 2.4 |
| Gen. chess | ✔ | 1.00000 | 1.00000 | 1.00000 | 5.2 |

Table 7: Perfect information games, refinement via conversation.

| Game | OOD | transition accuracy | | | # LLM calls |
| --- | --- | --- | --- | --- | --- |
| | | train | test | online | |
| Backgammon | ✗ | 1.00000 | 0.99944 | 1.00000 | 13.2 |
| Connect four | ✗ | 1.00000 | 1.00000 | 1.00000 | 3.2 |
| Tic-tac-toe | ✗ | 1.00000 | 1.00000 | 1.00000 | 2.0 |
| Gen. tic-tac-toe | ✔ | 1.00000 | 1.00000 | 1.00000 | 2.4 |
| Gen. chess | ✔ | 1.00000 | 1.00000 | 1.00000 | 4.2 |

Table 8: Imperfect info. games, refinement via tree search, hidden state inference.

| Game | OOD | transition accuracy | | | inference accuracy | | | # LLM calls |
| --- | --- | --- | --- | --- | --- | --- | --- | --- |
| | | train | test | online | train | test | online | |
| Bargaining | ✗ | 1.0000 | 0.9482 | 0.8712 | 1.0000 | 1.0000 | 1.0000 | 32.8 |
| Leduc poker | ✗ | 1.0000 | 0.9854 | 0.9942 | 1.0000 | 1.0000 | 1.0000 | 4.2 |
| Gin rummy | ✗ | 0.8943 | 0.8243 | 0.8909 | 1.0000 | 0.9513 | 0.9738 | 500.0 |
| Quadranto | ✔ | 1.0000 | 1.0000 | 0.9991 | 1.0000 | 0.9911 | 0.9876 | 7.4 |
| Hand of war | ✔ | 1.0000 | 0.9782 | 0.9806 | 1.0000 | 1.0000 | 1.0000 | 28.0 |

### D.1.2   ACCURACY OF LEARNED TRANSITION AND INFERENCE FUNCTIONS VS NUMBER OF LLM CALLS

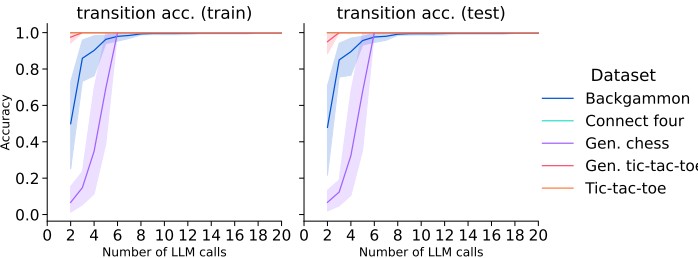

Figure 6: Evolution of the transition accuracy of the best generated CWM with the number of LLM calls for perfect games (with CWM refinement via tree search).

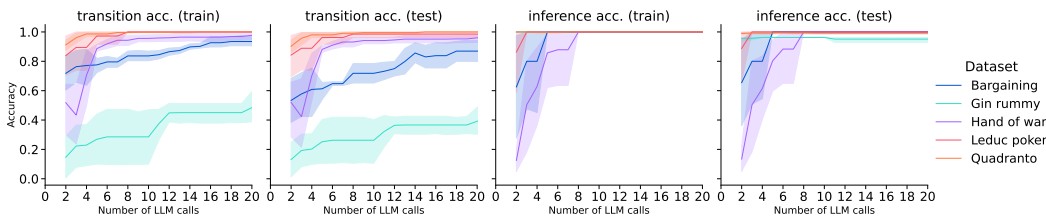

Figure 7: Evolution of the transition and inference accuracy with the number of LLM calls for imperfect games with refinement via tree search and hidden state inference.

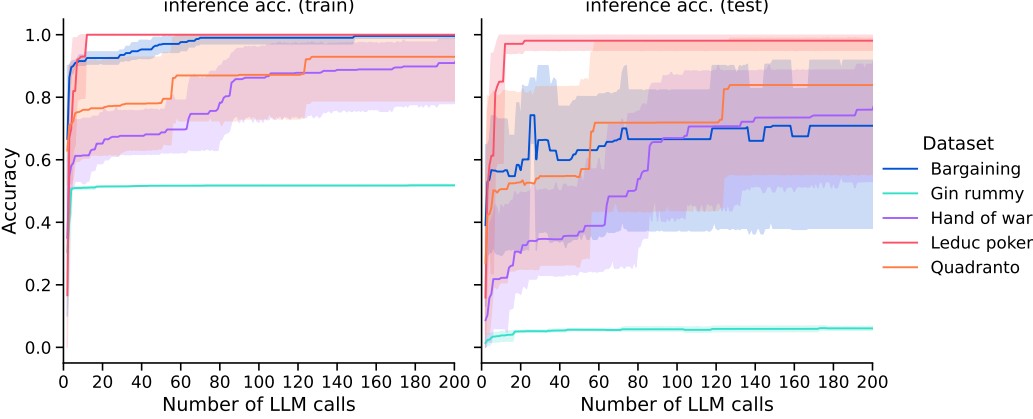

Figure 8: Evolution of the inference accuracy with the number of LLM calls for imperfect games with refinement via tree search with closed deck.

### D.1.3 TREE SEARCH SETTINGS

We use the following settings for treesearch throughout our experiments.

- heuristic_weight=5.0: Weight on the heuristic value (higher means more exploitation). The heuristic weight $C$ adjusts the parameters $\alpha$ and $\beta$ of the **Beta** prior on each arm Tang et al. (2024b). In particular, we set $\alpha = 1 + C \times h$ and $\beta = 1 + (1 - C) \times h$, where the heuristic value $h$ is the average pass rate of the unit tests.
- num_retries=500: Number of retries for tree search.
- num_tests_on_init=5: Number of tests of each type to include on the first synthesis.
- num_tests_on_error=1: Number of failed tests of each type to include during code refinement.
- min_heuristic_value_on_init=0.01: Minimum heuristic value to consider a node for expansion on initialization.
- min_heuristic_value_gain=0.01: Minimum heuristic value gain to consider a node for expansion.

## D.2 Detailed per-game arena results

For games in which the outcomes are win, lose or draw, we show the frequency of these 3 outcomes in 3 different columns, for each agent. For games with arbitrary payoff (Bargaining, LeDuc-Poker, Gin Rummy), we show the payoff to each player in 2 different columns. We consider the case when our agent acts as Player 0 or Player 1, and show these in different rows, to account for first-mover advantage.

For imperfect information games, we show results for hidden history inference (open deck learning), hidden state inference (open deck learning), and hidden history inference (closed deck learning).

For games in which the outcomes are win, lose or draw, we also report (in small font) the number of games with outcome that were forfeited vs the total number of games with that outcome. (A forfeit means the agent has either thrown an exception or tried to execute an illegal action, since our game arena API does not allow the agent to see which actions are legal at a given point in the game.)

### D.2.1 Perfect information games

Table 9: Win rates using CWM refinement via tree search against multiple opponents. For each game, results in the first (second) row correspond to our agent going first (second).

| Game | P | Gemini 2.5 Pro | | | GT-MCTS | | | Random | | |
|---|---|---|---|---|---|---|---|---|---|---|
| | | Win (forfeit/n) | Loss (forfeit/n) | Draw (n) | Win | Loss | Draw | Win | Loss | Draw |
| Backgammon | ○ | 1.00 (100/100) | 0.00 (0/0) | 0.00 (0) | 0.08 | 0.92 | 0.00 | 0.98 | 0.02 | 0.00 |
| | ● | 1.00 (100/100) | 0.00 (0/0) | 0.00 (0) | 0.07 | 0.93 | 0.00 | 0.98 | 0.02 | 0.00 |
| Connect four | ● | 1.00 (2/100) | 0.00 (0/0) | 0.00 (0) | 0.69 | 0.31 | 0.00 | 1.00 | 0.00 | 0.00 |
| | ● | 1.00 (1/100) | 0.00 (0/0) | 0.00 (0) | 0.28 | 0.72 | 0.00 | 1.00 | 0.00 | 0.00 |
| Tic-tac-toe | × | 0.05 (0/5) | 0.00 (0/0) | 0.95 (95) | 0.00 | 0.00 | 1.00 | 0.97 | 0.00 | 0.03 |
| | ○ | 0.00 (0/0) | 0.00 (0/0) | 1.00 (100) | 0.00 | 0.00 | 1.00 | 0.75 | 0.00 | 0.25 |
| Gen. tic-tac-toe | × | 0.89 (0/89) | 0.10 (0/10) | 0.01 (1) | 0.88 | 0.12 | 0.00 | 1.00 | 0.00 | 0.00 |
| | ○ | 0.93 (0/93) | 0.07 (0/7) | 0.00 (0) | 0.37 | 0.63 | 0.00 | 1.00 | 0.00 | 0.00 |
| Gen. chess | ♙ | 1.00 (92/100) | 0.00 (0/0) | 0.00 (0) | 0.18 | 0.43 | 0.39 | 1.00 | 0.00 | 0.00 |
| | ♟ | 1.00 (97/100) | 0.00 (0/0) | 0.00 (0) | 0.49 | 0.17 | 0.34 | 1.00 | 0.00 | 0.00 |

### D.2.2 HIDDEN HISTORY INFERENCE

Table 10: Payoffs using CWM refinement via tree search and hidden history inference against multiple opponents. For each game, results in the first (second) row correspond to our agent going first (second).

| Game | P | Gemini 2.5 Pro | | GT-ISMCTS | | Random | |
|---|---|---|---|---|---|---|---|
| | | Us | Them | Us | Them | Us | Them |
| Bargaining | 🏙 | 8.90 | 3.31 | 8.01 | 5.26 | 8.21 | 2.41 |
| | 🏘 | 8.80 | 4.73 | 7.51 | 5.44 | 8.12 | 2.81 |
| Leduc poker | ♦ | -0.03 | 0.03 | -0.65 | 0.65 | 0.86 | -0.86 |
| | ♠ | 1.55 | -1.55 | 0.24 | -0.24 | 1.09 | -1.09 |
| Gin rummy | ⬜ | 120.54 | -120.54 | -115.62 | 115.62 | -4.92 | 4.92 |
| | ⬛ | 123.00 | -123.00 | -115.62 | 115.62 | -15.99 | 15.99 |

Table 11: Win rates using CWM refinement via tree search and hidden history inference against multiple opponents. For each game, results in the first (second) row correspond to our agent going first (second).

| Game | P | Gemini 2.5 Pro | | | GT-ISMCTS | | | Random | | |
|---|---|---|---|---|---|---|---|---|---|---|
| | | Win (forfeit/n) | Loss (forfeit/n) | Draw (n) | Win | Loss | Draw | Win | Loss | Draw |
| Quadranto | ⬜ | 0.91 (16/91) | 0.08 (0/8) | 0.01 (1) | 0.06 | 0.02 | 0.92 | 0.27 | 0.03 | 0.70 |
| | ⬛ | 0.75 (0/75) | 0.18 (0/18) | 0.07 (7) | 0.11 | 0.08 | 0.81 | 0.31 | 0.06 | 0.63 |
| Hand of war | ♥ | 0.35 (16/35) | 0.56 (11/56) | 0.09 (9) | 0.20 | 0.72 | 0.08 | 0.33 | 0.57 | 0.10 |
| | ♣ | 0.33 (0/33) | 0.62 (39/62) | 0.05 (5) | 0.31 | 0.61 | 0.08 | 0.33 | 0.62 | 0.05 |

### D.2.3  HIDDEN STATE INFERENCE

Table 12: Payoffs using CWM refinement via tree search and hidden state inference against multiple opponents. For each game, results in the first (second) row correspond to our agent going first (second).

| Game | P | Gemini 2.5 Pro | | GT-ISMCTS | | Random | |
|---|---|---|---|---|---|---|---|
| | | Us | Them | Us | Them | Us | Them |
| Bargaining | 📚 | 8.48 | 4.32 | 7.46 | 4.17 | 7.70 | 2.72 |
| | 📙 | 7.98 | 6.42 | 7.25 | 6.04 | 7.78 | 3.47 |
| Leduc poker | ♦ | 1.75 | -1.75 | 0.16 | -0.16 | 1.19 | -1.19 |
| | ♠ | 0.37 | -0.37 | 0.41 | -0.41 | 1.12 | -1.12 |
| Gin rummy | ◻ | 66.42 | -66.42 | -114.39 | 114.39 | -28.29 | 28.29 |
| | ◼ | 121.77 | -121.77 | -121.77 | 121.77 | -4.92 | 4.92 |

Table 13: Win rates using CWM refinement via tree search and hidden state inference against multiple opponents. For each game, results in the first (second) row correspond to our agent going first (second).

| Game | P | Gemini 2.5 Pro | | | GT-ISMCTS | | | Random | | |
|---|---|---|---|---|---|---|---|---|---|---|
| | | Win (forfeit/n) | Loss (forfeit/n) | Draw (n) | Win | Loss | Draw | Win | Loss | Draw |
| Quadranto | ◻ | 0.58 (16/58) | 0.40 (0/40) | 0.02 (2) | 0.13 | 0.02 | 0.85 | 0.19 | 0.04 | 0.77 |
| | ● | 0.37 (0/37) | 0.54 (0/54) | 0.09 (9) | 0.14 | 0.01 | 0.85 | 0.26 | 0.07 | 0.67 |
| Hand of war | ♥ | 0.41 (16/41) | 0.49 (0/49) | 0.10 (10) | 0.25 | 0.61 | 0.14 | 0.59 | 0.28 | 0.13 |
| | ♣ | 0.44 (0/44) | 0.40 (0/40) | 0.16 (16) | 0.42 | 0.41 | 0.17 | 0.63 | 0.24 | 0.13 |

### D.2.4 HIDDEN HISTORY INFERENCE WITH CLOSED DECK LEARNING

Table 14: Payoffs using CWM refinement via tree search with closed deck against multiple opponents. For each game, results in the first (second) row correspond to our agent going first (second).

| Game | P | Gemini 2.5 Pro | | GT-ISMCTS | | Random | |
|---|---|---|---|---|---|---|---|
| | | Us | Them | Us | Them | Us | Them |
| Bargaining | 🟩 | 7.03 | 5.91 | 7.01 | 5.98 | 7.37 | 3.76 |
| | 🟦 | 7.07 | 7.02 | 6.83 | 6.01 | 7.39 | 3.68 |
| Leduc poker | ♦ | 0.57 | -0.57 | -0.56 | 0.56 | 0.86 | -0.86 |
| | ♠ | 0.49 | -0.49 | -0.21 | 0.21 | 1.71 | -1.71 |
| Gin rummy | ⬜ | 29.52 | -29.52 | -114.39 | 114.39 | -119.31 | 119.31 |
| | ⬛ | -63.96 | 63.96 | -119.31 | 119.31 | -121.77 | 121.77 |

Table 15: Win rates using CWM refinement via tree search with closed deck against multiple opponents. For each game, results in the first (second) row correspond to our agent going first (second).

| Game | P | Gemini 2.5 Pro | | | GT-ISMCTS | | | Random | | |
|---|---|---|---|---|---|---|---|---|---|---|
| | | Win (forfeit/n) | Loss (forfeit/n) | Draw (n) | Win | Loss | Draw | Win | Loss | Draw |
| Quadranto | ⬜ | 0.69 (0/69) | 0.26 (0/26) | 0.05 (5) | 0.08 | 0.05 | 0.87 | 0.23 | 0.05 | 0.72 |
| | ⬛ | 0.71 (2/71) | 0.22 (0/22) | 0.07 (7) | 0.13 | 0.09 | 0.78 | 0.27 | 0.05 | 0.68 |
| Hand of war | ♥ | 0.41 (16/41) | 0.42 (0/42) | 0.17 (17) | 0.31 | 0.57 | 0.12 | 0.61 | 0.24 | 0.15 |
| | ♣ | 0.54 (0/54) | 0.25 (0/25) | 0.21 (21) | 0.47 | 0.40 | 0.13 | 0.62 | 0.26 | 0.12 |

## D.3 FORFEIT RATES FOR NON-TERNARY-OUTCOME GAMES

Table 16: Forfeit rates for non-ternary-outcome games using CWM refinement via tree search and hidden history inference against multiple opponents. This is the rate at which each agent forfeits the game by failing to execute a legal action. For each game, results in the first (second) row correspond to our agent going first (second).

| Game | P | Gemini 2.5 Pro | | GT-ISMCTS | | Random | |
| --- | --- | --- | --- | --- | --- | --- | --- |
| | | Us | Them | Us | Them | Us | Them |
| Bargaining | | 0.00 | 0.00 | 0.00 | 0.00 | 0.00 | 0.00 |
| | | 0.00 | 0.01 | 0.00 | 0.00 | 0.00 | 0.00 |
| Leduc poker | ♦ | 0.00 | 0.00 | 0.00 | 0.00 | 0.00 | 0.00 |
| | ♠ | 0.00 | 0.00 | 0.00 | 0.00 | 0.00 | 0.00 |
| Gin rummy | | 0.01 | 0.99 | 0.94 | 0.00 | 0.04 | 0.00 |
| | | 0.00 | 1.00 | 0.94 | 0.00 | 0.13 | 0.00 |

Table 17: Forfeit rates for non-ternary-outcome games using CWM refinement via tree search and hidden state inference against multiple opponents. This is the rate at which each agent forfeits the game by failing to execute a legal action. For each game, results in the first (second) row correspond to our agent going first (second).

| Game | P | Gemini 2.5 Pro | | GT-ISMCTS | | Random | |
| --- | --- | --- | --- | --- | --- | --- | --- |
| | | Us | Them | Us | Them | Us | Them |
| Bargaining | | 0.00 | 0.00 | 0.00 | 0.00 | 0.00 | 0.00 |
| | | 0.00 | 0.00 | 0.00 | 0.00 | 0.00 | 0.00 |
| Leduc poker | ♦ | 0.00 | 0.16 | 0.00 | 0.00 | 0.00 | 0.00 |
| | ♠ | 0.00 | 0.00 | 0.00 | 0.00 | 0.00 | 0.00 |
| Gin rummy | | 0.23 | 0.77 | 0.93 | 0.00 | 0.23 | 0.00 |
| | | 0.00 | 0.99 | 0.99 | 0.00 | 0.04 | 0.00 |

Table 18: Forfeit rates for non-ternary-outcome games using CWM refinement via tree search with closed deck against multiple opponents. This is the rate at which each agent forfeits the game by failing to execute a legal action. For each game, results in the first (second) row correspond to our agent going first (second).

| Game | P | Gemini 2.5 Pro | | GT-ISMCTS | | Random | |
| --- | --- | --- | --- | --- | --- | --- | --- |
| | | Us | Them | Us | Them | Us | Them |
| Bargaining | | 0.00 | 0.00 | 0.00 | 0.00 | 0.00 | 0.00 |
| | | 0.00 | 0.00 | 0.00 | 0.00 | 0.00 | 0.00 |
| Leduc poker | ♦ | 0.00 | 0.00 | 0.00 | 0.00 | 0.00 | 0.00 |
| | ♠ | 0.00 | 0.00 | 0.00 | 0.00 | 0.00 | 0.00 |
| Gin rummy | | 0.38 | 0.62 | 0.93 | 0.00 | 0.97 | 0.00 |
| | | 0.76 | 0.24 | 0.97 | 0.00 | 0.99 | 0.00 |

### D.4 VALUE FUNCTION ABLATIONS

As explained in the main text, the purpose of value functions is to speed up (IS)MCTS by providing a better value initialization for new leaf nodes. This can also result in higher quality selections for a fixed budget. Synthetic value functions are generated by the LLM in one-shot, and its usefulness assessed via a tournament ran on top of the synthesized CWM. Agents using different value functions (or potentially no value function) compete against each other the synthesized CWM to evaluate performance.

The use of value function only delivered improvements in the case of Gen. tic-tac-toe and Bargaining, so our agent only used value functions when playing those games. Note that the choice to use value functions or not can be assessed before actual online game play, by having the agent play locally (with and without using a value function) on its own synthetic CWM as a proxy, and assessing which option is most beneficial.

Fig. 9 shows the ablation corresponding to not using a value function in Gen. tic-tac-toe and Bargaining.

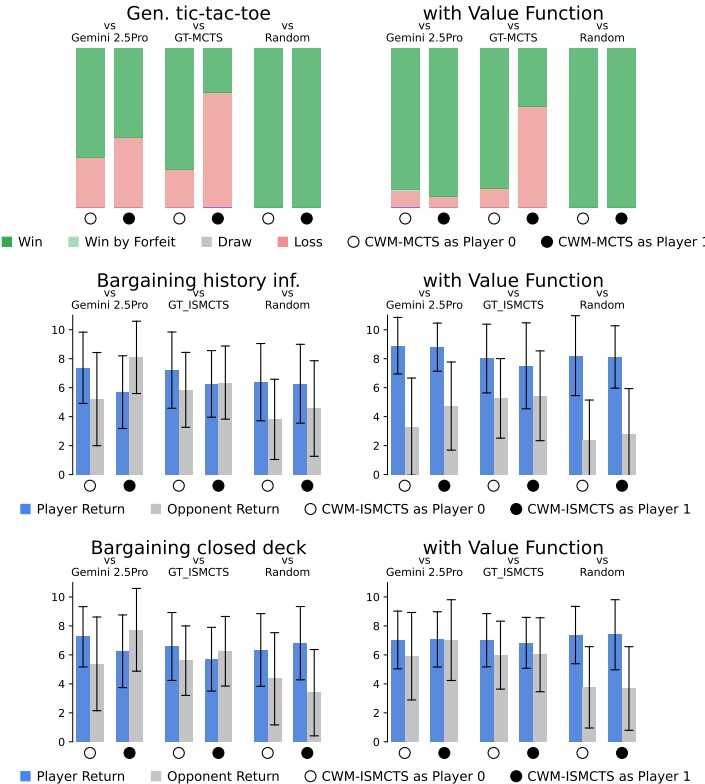

Figure 9: Ablation for Gen. tic-tac-toe and Bargaining. Effect of using synthesized value functions (right column) vs not (left column) to improve planning in CWMs.

# E PLANNING WITH PPO INSTEAD OF (IS)MCTS

## E.1 TRAINING A PPO AGENT ON TOP OF A CWM

The CWM agent discussed in the main paper relies on (IS)MCTS to take actions within its learned CWM. While effective, this online planning process can be slow. We investigate an alternative approach: amortizing the planning computation into a reactive policy, trained with the PPO algorithm (Schulman et al., 2017).

We entirely learn this PPO policy within the learned CWM environment. For each game, we train a PPO-CWM agent (acting either as Player 0 or Player 1) to maximize its rewards against an opponent that uniformly picks a legal action.

**Mapping JSON observations to 1D tensors:** The CWM represents observations in the JSON format provided by OpenSpiel, whereas the actor-critic networks we use (which are based on MLPs and RNNs) requires fixed-size 1D arrays as input.[6] Consequently, we need a procedure to map each JSON into a flat tensor representation. We generate this mapping programmatically by prompting a LLM as shown below, providing the CWM training sequences as examples.

```
You are an expert reinforcement learning researcher and Python programmer.

Your task is to implement the following two functions which form a bijective pair:

def observation_to_tensor(obs) -> np.ndarray: # 1D
  ...

def tensor_to_observation(tensor) -> np.ndarray: # 1D
  ...

An example input dataset is as follows:
{example}

First reason about the problem and possible corner cases. Finally output only
the resulting two functions without any placeholders.
```

**Architecture.** Our PPO agent uses an actor-critic architecture. For perfect information games, the actor and critic networks share a common feature extractor consisting of two 256-unit fully-connected layers with tanh activations. The actor head is a final linear layer that outputs logits for each action, which are then masked to ensure only legal moves are considered. The critic head is a separate linear layer that outputs a single scalar value.

For imperfect information games, we augment this architecture with a recurrent neural network to process historical information. An input observation $x_t$ is first passed through a 256-unit linear layer (with tanh activation). The result is fed into an RNN along with the previous hidden state $h_{t-1}$ to produce an output vector. This output is concatenated with the original input $x_t$ and passed through a final 256-unit hidden layer (with tanh activation) before being fed to the actor and critic heads as described above.

**PPO training.** The PPO-CWM agent is trained for a total of 10M agent steps inside the CWM, using the hyperparameters in Table 19. From the two-player trajectories collected, we extract the single-agent sequence of observations, actions, and rewards corresponding to the PPO-CWM agent. This filtered data is used to compute the advantages and the final PPO loss objective.

For each game, and each player, we train 5 PPO-CWM agents with different seeds, and select the one with the highest win rate against the random opponent for final evaluation. This agent is then benchmarked in the Arena, as described in Sec. 5.2. We include matches against our CWM MCTS agent to compare both approaches for leveraging the CWM.

---

[6]We could use transformers, which can handle JSON strings, for the actor and critic, but such models would be much slower to train.

Table 19: PPO hyperpameters.

| Module | Hyperparameter | Value |
|---|---|---|
| Environment | Number of environments | 50 |
| | Rollout horizon in environment | 100 |
| Advantage | $\gamma$ | 0.99 |
| | $\lambda$ | 0.95 |
| Loss | $\epsilon$ clipping | 0.2 |
| | Value loss coefficient | 0.5 |
| | Entropy loss coefficient start | 0.1 |
| | Entropy loss coefficient end | 0.01 |
| | Entropy loss coefficient schedule | Linear |
| Learning | Optimizer | Adam (Kingma & Ba, 2014) |
| | Learning rate | 0.0003 |
| | Max. gradient norm | 0.5 |
| | Learning rate annealing | False |
| | Number of minibatches (MFRL) | 10 |
| | Number of epochs (MFRL) | 4 |

## E.2 RESULTS

Arena results are presented in Tables 20 to 24. Note that PPO-CWM was not trained on Gin Rummy due to the poor performance of the CWM on that game.

**PPO-CWM vs. Random.** PPO-CWM outperforms the random agent for all the games.

**PPO-CWM vs. Gemini 2.5 Pro.** Our PPO-CWM agent outperforms or matches Gemini in all the games. For perfect information games, PPO-CWM wins in Backgammon, Generalized Chess and Tic-Tac-Toe; and exhibits mixed results (winning as one player and losing as the other) in Connect Four and Generalized Tic-Tac-Toe. For imperfect information games, for both open deck and closed deck, PPO-CWM wins in Bargaining and Quadranto, and ties in Hand of War and Leduc Poker.

**PPO-CWM vs. CWM MCTS.** For perfect information games, where the learned CWM is a near-perfect replica of the environment, PPO-CWM is outperformed by our CWM-MCTS agent. The only exception is Generalized Tic-Tac-Toe when PPO-CWM acts as Player 0. For imperfect information games, PPO-CWM wins in two games (Hand of War and Bargaining) and loses in the other two games (Leduc poker and Quadranto).

### E.2.1 GAMES WITH PERFECT INFORMATION

Table 20: PPO-CWM win rates using CWM refinement via tree search against multiple opponents. For each game, results in the first (second) row correspond to our agent going first (second).

| Game | P | CWM MCTS | | | Gemini 2.5 Pro | | | GT-MCTS | | | Random | | |
|---|---|---|---|---|---|---|---|---|---|---|---|---|---|
| | | Win (forfeit/n) | Loss (forfeit/n) | Draw (n) | Win (forfeit/n) | Loss (forfeit/n) | Draw (n) | Win | Loss | Draw | Win | Loss | Draw |
| Backgammon | ○ | 0.01 (0/1) | 0.99 (0/99) | 0.00 (0) | 1.00 (100/100) | 0.00 (0/0) | 0.00 (0) | 0.02 | 0.98 | 0.00 | 0.92 | 0.08 | 0.00 |
| | ● | 0.03 (0/3) | 0.97 (0/97) | 0.00 (0) | 1.00 (100/100) | 0.00 (0/0) | 0.00 (0) | 0.01 | 0.99 | 0.00 | 0.94 | 0.06 | 0.00 |
| Connect four | ● | 0.00 (0/0) | 1.00 (0/100) | 0.00 (0) | 0.92 (0/92) | 0.08 (0/8) | 0.00 (0) | 0.00 | 1.00 | 0.00 | 1.00 | 0.00 | 0.00 |
| | ● | 0.00 (0/0) | 1.00 (0/100) | 0.00 (0) | 0.02 (0/2) | 0.98 (0/98) | 0.00 (0) | 0.00 | 1.00 | 0.00 | 0.99 | 0.01 | 0.00 |
| Tic-tac-toe | ✕ | 0.00 (0/0) | 0.00 (0/0) | 1.00 (100) | 0.00 (0/0) | 0.00 (0/0) | 1.00 (100) | 0.00 | 0.00 | 1.00 | 1.00 | 0.00 | 0.00 |
| | ○ | 0.00 (0/0) | 1.00 (0/100) | 0.00 (0) | 0.87 (0/87) | 0.12 (0/12) | 0.01 (1) | 0.00 | 1.00 | 0.00 | 0.91 | 0.01 | 0.08 |
| Gen. tic-tac-toe | ✕ | 0.45 (0/45) | 0.55 (0/55) | 0.00 (0) | 0.91 (0/91) | 0.09 (0/9) | 0.00 (0) | 0.54 | 0.46 | 0.00 | 1.00 | 0.00 | 0.00 |
| | ○ | 0.04 (0/4) | 0.96 (0/96) | 0.00 (0) | 0.38 (0/38) | 0.62 (0/62) | 0.00 (0) | 0.05 | 0.95 | 0.00 | 0.99 | 0.01 | 0.00 |
| Gen. chess | ♙ | 0.00 (0/0) | 1.00 (0/100) | 0.00 (0) | 0.94 (90/94) | 0.06 (0/6) | 0.00 (0) | 0.00 | 1.00 | 0.00 | 1.00 | 0.00 | 0.00 |
| | ♟ | 0.08 (0/8) | 0.92 (0/92) | 0.00 (0) | 0.95 (5/95) | 0.05 (0/5) | 0.00 (0) | 0.08 | 0.92 | 0.00 | 1.00 | 0.00 | 0.00 |

### E.2.2 HIDDEN HISTORY INFERENCE

Table 21: PPO-CWM win rates using CWM refinement via tree search and hidden history inference against multiple opponents. For each game, results in the first (second) row correspond to our agent going first (second).

| Game | P | CWM MCTS | | | Gemini 2.5 Pro | | | GT-ISMCTS | | | Random | | |
|------|---|----------|---|---|----------------|---|---|-----------|---|---|--------|---|---|
| | | Win (forfeit/n) | Loss (forfeit/n) | Draw (n) | Win (forfeit/n) | Loss (forfeit/n) | Draw (n) | Win | Loss | Draw | Win | Loss | Draw |
| Quadranto | ○ | 0.01 (1/1) | 0.64 (0/64) | 0.35 (35) | 0.61 (0/61) | 0.19 (0/19) | 0.20 (20) | 0.00 | 0.62 | 0.38 | 0.51 | 0.18 | 0.31 |
| | ● | 0.02 (0/2) | 0.30 (0/30) | 0.68 (68) | 0.64 (0/64) | 0.25 (0/25) | 0.11 (11) | 0.01 | 0.46 | 0.53 | 0.43 | 0.13 | 0.44 |
| Hand of war | ♥ | 0.75 (41/75) | 0.19 (0/19) | 0.06 (6) | 0.30 (0/30) | 0.47 (0/47) | 0.23 (23) | 0.32 | 0.58 | 0.10 | 0.66 | 0.19 | 0.15 |
| | ♣ | 0.64 (35/64) | 0.29 (0/29) | 0.07 (7) | 0.54 (0/54) | 0.29 (0/29) | 0.17 (17) | 0.38 | 0.56 | 0.06 | 0.66 | 0.19 | 0.15 |

Table 22: PPO-CWM payoffs using CWM refinement via tree search and hidden history inference against multiple opponents. For each game, results in the first (second) row correspond to our agent going first (second).

| Game | P | CWM MCTS | | Gemini 2.5 Pro | | GT-ISMCTS | | Random | |
|------|---|----------|---|----------------|---|-----------|---|--------|---|
| | | Us | Them | Us | Them | Us | Them | Us | Them |
| Bargaining | 🟩 | 7.92 | 4.53 | 8.69 | 4.22 | 7.92 | 4.31 | 7.77 | 2.72 |
| | 🟦 | 7.57 | 4.53 | 8.37 | 5.11 | 8.04 | 4.88 | 7.87 | 3.24 |
| Leduc poker | ♦ | -0.60 | 0.60 | -0.82 | 0.82 | -0.54 | 0.54 | 1.25 | -1.25 |
| | ♠ | -1.58 | 1.58 | 0.95 | -0.95 | -2.26 | 2.26 | 1.99 | -1.99 |

### E.2.3 HIDDEN STATE INFERENCE

Table 23: PPO-CWM win rates using CWM refinement via tree search and hidden state inference against multiple opponents. For each game, results in the first (second) row correspond to our agent going first (second).

| Game | P | CWM MCTS | | | Gemini 2.5 Pro | | | GT-ISMCTS | | | Random | | |
|------|---|----------|---|---|----------------|---|---|-----------|---|---|--------|---|---|
| | | Win (forfeit/n) | Loss (forfeit/n) | Draw (n) | Win (forfeit/n) | Loss (forfeit/n) | Draw (n) | Win | Loss | Draw | Win | Loss | Draw |
| Quadranto | ○ | 0.01 (0/1) | 0.54 (0/54) | 0.45 (45) | 0.75 (0/75) | 0.18 (0/18) | 0.07 (7) | 0.03 | 0.52 | 0.45 | 0.59 | 0.11 | 0.30 |
| | ● | 0.18 (0/18) | 0.48 (0/48) | 0.34 (34) | 0.74 (0/74) | 0.25 (0/25) | 0.01 (1) | 0.00 | 0.72 | 0.28 | 0.57 | 0.14 | 0.29 |
| Hand of war | ♥ | 0.40 (0/40) | 0.47 (0/47) | 0.13 (13) | 0.35 (0/35) | 0.55 (0/55) | 0.10 (10) | 0.27 | 0.63 | 0.10 | 0.60 | 0.26 | 0.14 |
| | ♣ | 0.53 (0/53) | 0.35 (0/35) | 0.12 (12) | 0.57 (0/57) | 0.24 (0/24) | 0.19 (19) | 0.50 | 0.44 | 0.06 | 0.53 | 0.30 | 0.17 |

Table 24: PPO-CWM payoffs using CWM refinement via tree search and hidden state inference against multiple opponents. For each game, results in the first (second) row correspond to our agent going first (second).

| Game | P | CWM MCTS | | Gemini 2.5 Pro | | GT-ISMCTS | | Random | |
|------|---|----------|---|----------------|---|-----------|---|--------|---|
| | | Us | Them | Us | Them | Us | Them | Us | Them |
| Bargaining | 🟩 | 8.45 | 3.93 | 8.32 | 4.76 | 8.53 | 4.83 | 8.00 | 2.64 |
| | 🟦 | 6.44 | 3.80 | 7.98 | 4.66 | 7.82 | 4.60 | 8.21 | 2.67 |
| Leduc poker | ♦ | -1.93 | 1.93 | 0.31 | -0.31 | -1.56 | 1.56 | 1.58 | -1.58 |
| | ♠ | -0.51 | 0.51 | -0.57 | 0.57 | -1.29 | 1.29 | 1.99 | -1.99 |

### E.2.4 HIDDEN HISTORY INFERENCE WITH CLOSED DECK LEARNING

Table 25: PPO-CWM win rates using CWM refinement via tree search with closed deck against multiple opponents. For each game, results in the first (second) row correspond to our agent going first (second).

| Game | P | CWM MCTS | | | Gemini 2.5 Pro | | | GT-ISMCTS | | | Random | | |
|---|---|---|---|---|---|---|---|---|---|---|---|---|---|
| | | Win (forfeit/n) | Loss (forfeit/n) | Draw (n) | Win (forfeit/n) | Loss (forfeit/n) | Draw (n) | Win | Loss | Draw | Win | Loss | Draw |
| Quadranto | ○ | 0.01 (0/1) | 0.59 (0/59) | 0.40 (40) | 0.74 (2/74) | 0.24 (0/24) | 0.02 (2) | 0.01 | 0.60 | 0.39 | 0.51 | 0.11 | 0.38 |
| | ● | 0.02 (0/2) | 0.47 (0/47) | 0.51 (51) | 0.70 (0/70) | 0.27 (0/27) | 0.03 (3) | 0.00 | 0.37 | 0.63 | 0.58 | 0.16 | 0.26 |
| Hand of war | ♥ | 0.50 (0/50) | 0.41 (0/41) | 0.09 (9) | 0.32 (0/32) | 0.49 (0/49) | 0.19 (19) | 0.27 | 0.64 | 0.09 | 0.58 | 0.26 | 0.16 |
| | ♣ | 0.51 (0/51) | 0.38 (0/38) | 0.11 (11) | 0.47 (0/47) | 0.30 (0/30) | 0.23 (23) | 0.46 | 0.45 | 0.09 | 0.70 | 0.20 | 0.10 |

Table 26: PPO-CWM payoffs using CWM refinement via tree search with closed deck against multiple opponents. For each game, results in the first (second) row correspond to our agent going first (second).

| Game | P | CWM MCTS | | Gemini 2.5 Pro | | GT-ISMCTS | | Random | |
|---|---|---|---|---|---|---|---|---|---|
| | | Us | Them | Us | Them | Us | Them | Us | Them |
| Bargaining | 🟩 | 5.96 | 5.90 | 7.15 | 4.96 | 6.35 | 5.21 | 7.07 | 3.21 |
| | 🟦 | 6.70 | 5.49 | 7.73 | 5.10 | 7.40 | 5.34 | 8.07 | 2.79 |
| Leduc poker | ♦ | -1.69 | 1.69 | -0.63 | 0.63 | -1.05 | 1.05 | 1.72 | -1.72 |
| | ♠ | -0.08 | 0.08 | 1.61 | -1.61 | -1.78 | 1.78 | 2.25 | -2.25 |

## F    AUTOMATIC REJECTION OF BAD CWM SAMPLES

The CWM refinement process can occasionally produce a low-quality CWM. This is rarely the case for perfect information games, where more information is available for refinement and unit tests are more strict, but we have observed this happening in the case of imperfect information games. To reduce this effect, in the case of imperfect information games, we sample 5 CWMs, create a CWM-ISMCTS agent from each one, and make those agents compete against each other. Agents are then ranked according to the average payoff obtained in those competitions. Agents that are worse than the best scoring agent by more than 10% of the observed utility range are rejected.

Since we do not have access to the ground truth game for these competitions, the agents use the CWM of one of them as a stand-in for the actual game. We call the CWM used to play the game the *host*. This means that we have 2 possible hosts $\times$ 5 agents acting as Player $0 \times 5$ agents acting as Player 1. This results in a total of 50 possible matches. Since the outcome of a match is stochastic, we repeat each match 50 times. Execution failures or the execution of illegal actions during these games result in both players losing the game.

## G    SKETCH OF INFORMATION FLOW OF EACH AGENT

Here we provide a sketch of the information flow for each the agents. Of course, many details are omitted, and the prompts are highly simplified, see Appendix H for the actual prompts.

```python
def llm_agent_generator(LLM, rules, traj):
  prompt = (f"You are playing a game with these rules: {rules}.\n"
            f"Example trajectories: {traj}.\n")

  def policy(action_obs_history):
    return LLM(prompt + f"Action-observation history: {action_obs_history}. "
               "Pick the next best action.")
  return policy

def cwm_agent_perfect_info_generator(LLM, rules, traj, GT=False):
  M = induce_cwm(LLM, rules, traj) if not GT else ground_truth_M
  V = induce_value_fn(LLM, rules, traj, M)

  def policy(action_obs_history):
    return MCTS(action_obs_history[-1], M, V)
  return policy

def cwm_agent_imperfect_info_generator(LLM, rules, traj, GT=False):
  (M, I) = induce_cwm_pomdp(LLM, rules, traj) if not GT else ground_truth_MI
  V = induce_value_fn(LLM, rules, traj, M)
  def policy(action_obs_history):
    return ISMCTS(action_obs_history, M, V, I)
  return policy

def induce_cwm_zero_shot(LLM, rules, traj):
  prompt = (f"You are playing a game with these rules: {rules}.\n"
            f"Generate python code that matches this API: {fn_signature}\n"
            f"The code should pass these unit tests: {make_tests(traj)}\n")
  return LLM(prompt)
```

## H System and agent prompts

### H.1 Tree search

Our tree search prompt is:

```
You are an expert python programmer who is building the game of {game_name}.
Here is a description of the game:
{game_desc}

The goal is to implement a python function with the following signature.
# START FUNCTION SIGNATURE
{function_signature}
# END FUNCTION SIGNATURE

The original implementation is as follow. Please try to refine the original code.
# START CODE BLOCK
{orig_code}
# END CODE BLOCK

Your code should satisfy the following unit tests.
Your code should fix the TODO errors in the comments of the unit tests, if any.
# START UNIT TESTS
{test_code}
# END UNIT TESTS

Do not repeat the unit tests, only return the functions.
Do not leave placeholders.

Do not repeat the function signature.
Do not copy the unit tests.

Only produce code that is compact.
Do write comments explaining what the code does.
Do use helper functions to reduce code duplication.

Start by reasoning about the game and the unit tests.
Also reason about the errors and possible fixes.

Finally, try to write {num_targets} versions of the code.
Make sure each code is in a different code blocks starting with ```python.
```

`function_signature` contains the function definition for the LLM to fill out, while `test_code` defines the properties (expressed as unit tests) that the resulting code needs to satisfy.

`function_signature` and `test_code` both depend on if the game is a perfect or imperfect information game, whether it is being learnt in an open or closed deck fashion, and if the inference is perform via hidden history or hidden state inference. These variations are defined in the following sections.

Finally, `orig_code` is the code being refined at each iteration. On the first iteration, this paragraph is not present.

### H.2 Perfect information games

`function_signature` is defined as follows:

```
Action: str
State: dict[str, Any]
PlayerObservation: dict[str, Any]
```

```python
def apply_action(state: State, action: Action) -> State:
  """Returns the new state after an action has been taken."""

def get_current_player(state: State) -> int:
  """Returns current player, with -1 for chance and -4 for terminal."""

def get_player_name(player_id: int) -> str:
  """Returns the name of the player, with 'chance' for -1, and 'terminal' for -4."""

def get_rewards(state: State) -> list[float]:
  """Returns the rewards per player from their last action."""

def get_legal_actions(state: State) -> list[Action]:
  """Returns legal actions that can be taken in current state."""

def get_observations(state: State) -> list[PlayerObservation]:
  """Returns the observation for player."""
```

`test_code` tests the transition between two states, testing each of the API calls defined in `function_signature`. Here is an example transition unit test for tic tac toe, where the board is provided as a flat 1D array:

```python
class TestTransition2(unittest.TestCase):
  def test_transition_2(self):
    state = {'board': [None, None, None, None, 'x', None, 'o', None, None], '
        current_player_mark': 'x'}

    self.assertEqual(0, get_current_player(state))
    self.assertEqual('0', get_player_name(0))
    self.assertEqual([0.0, 0.0], get_rewards(state))
    self.assertEqual([{'board': [None, None, None, None, 'x', None, 'o', None, None], '
        current_player_mark': 'x'}, {'board': [None, None, None, None, 'x', None, 'o',
        None, None], 'current_player_mark': 'x'}], get_observations(state))
    self.assertSetEqual(set(['x(0,0)', 'x(0,1)', 'x(0,2)', 'x(1,0)', 'x(1,2)', 'x(2,1)',
         'x(2,2)']), set(get_legal_actions(state)))
    self.assertEqual({'board': [None, None, None, 'x', 'x', None, 'o', None, None], '
        current_player_mark': 'o'}, apply_action(state, 'x(1,0)'))
```

If this test has failed, the LLM is provided with the python error message in the form of a comment before the test. The use of `self.assertEqual` style functions ensures that the LLM is provided with a rich description of how the expected and actual data structures vary.

### H.3 HIDDEN HISTORY INFERENCE FUNCTION SYNTHESIS, OPEN DECK

`function_signature` starts with the version from Section H.1, then adds the inference definition:

```python
def resample_history(
  obs_action_history: list[tuple[PlayerObservation, Action | None]],
  player_id: int
) -> list[Action]:
  """Stochastically sample one of many potential history of actions for all players(
      including 'chance' and 'terminal')

  This is given only a single player's observations and actions, and needs to recreate
      the player_id's observations

def resample_history(obs_action_history: list[tuple[PlayerObservation, Action | None]],
    player_id: int) -> list[Action]:
  """Stochastically sample one of many potential history of actions for all players(
      including 'chance' and 'terminal')

  This is given only a single player's observations and actions, and needs to recreate
      the player_id's observations
```

`unit_text` again starts with the definition from Section H.1 then adds the following test for added inference function:

```
state = INITIAL_STATE
obs_action_history = {obs_action_history}
obs_and_action_iter = iter(obs_action_history)
current_player_obs, current_player_action = next(obs_and_action_iter)
player_id = {player_id}
for action in resample_history(obs_action_history, player_id):
  print(f"In state {{state}}")
  if get_current_player(state) == player_id:
    self.assertEqual(current_player_obs, get_observations(state)[player_id])
    print(f"Recreated observation {{current_player_obs}}")
    self.assertEqual(current_player_action, action)
    current_player_obs, current_player_action = next(obs_and_action_iter)

  print(f"Taking action {{action}}")
  state = apply_action(state, action)
try:
  next(obs_and_action_iter)
  raise ValueError('Failed to iterate through all observations.')
except StopIteration:
  pass
self.assertEqual(player_id, get_current_player(state))
```

where `INITIAL_STATE` is provided at the beginning of the unit tests and is the static first state of the game. `obs_action_history` is the history of observations and actions for player `player_id` for which we want to resample the history of actions that lead to the current observations.

Note the presence of print statements inside the unit test. The last ten lines of standard output are provided to the LLM in addition to the error message.

## H.4 HIDDEN STATE INFERENCE FUNCTION SYNTHESIS

`function_signature` again starts with the version from Section H.1, then adds the inference function definition:

```
def resample_state(obs_action_history: list[tuple[PlayerObservation, Action | None]],
    player_id: int) -> list[int]:
  """Stochastically sample one of the reachable statess for player given the observation
      and action history that recreates the player's observation."""
```

`unit_test` again starts with the definition from Section H.1 then adds the following test for added inference function above:

```
obs_action_history = {obs_action_history}
player_id = {player_id}
resampled_state = resample_state(obs_action_history, player_id)

self.assertEqual(obs_action_history[-1][0], get_observations(resampled_state)[player_id
    ])
```

## H.5 HIDDEN HISTORY INFERENCE FUNCTION SYNTHESIS, CLOSED DECK

`function_signature` is similar to that in Section H.2:

```
def resample_history(obs_action_history: list[tuple[PlayerObservation, Action | None]],
    player_id: int, last_is_terminal: bool) -> list[Action]:
  """Stochastically sample one of many potential histories of actions for all players(
      including 'chance' and 'terminal')
  given only a single player's observations and actions.

  It needs to recreate the player_id's observations.
  last_is_terminal indicates if the last player observation is from end of game when
      player_id is -4."""
```

Note the extra argument `last_is_terminal`. This indicates that the final observation in `obs_action_history` is of the terminal state. This allows adding tests that resample the entire game from the beginning to the terminal state, testing the ability of the LLM to predict the final reward of the player. In open deck, the transition tests cover this. For simplicity, we assumed that the rewards are terminal but this is easy to relax.

The corresponding `unit_tests` for the inference function is:

```
state = INITIAL_STATE
obs_action_history = {obs_action_history}
player_id = {player_id}
last_is_terminal = {ends_in_terminal}
obs_and_action_iter = iter(obs_action_history)
current_player_obs, current_player_action = next(obs_and_action_iter)
for action in resample_history(obs_action_history, player_id, last_is_terminal):
  print(f"In state {{state}}")
  if get_current_player(state) == player_id:
    self.assertEqual(current_player_obs, get_observations(state)[player_id])
    print(f"Recreated observation {{current_player_obs}}")
    self.assertEqual(current_player_action, action)
    current_player_obs, current_player_action = next(obs_and_action_iter)

  print(f"Taking action {{action}}")
  state = apply_action(state, action)
try:
  next(obs_and_action_iter)
  raise ValueError('Failed to iterate through all observations.')
except StopIteration:
  pass
```

Again, this is very similar to `unit_test` in Section H.2, but also covers the terminal state of the game and it's associated reward.

Note that no transition unit tests are added as we do not have access to the state. However, just testing the inference function is not enough to ensure that the resulting closed deck game is playable. Instead, a random play test is added to `unit_test`:

```
state = {initial_state}
rg = np.random.RandomState({seed})
for it in range(1000): # upper bound on game length
  current_player = get_current_player(state)
  rewards = get_rewards(state)
  assert len(rewards) == 2
  print (f"State is {{state}}, current player is {{current_player}}, rewards are {{
      rewards}}")

  if current_player == -4: # Game over
    break
  if current_player in [0,1]: # Real players
    print(f"Observation for current player is {{get_observations(state)[current_player
        ]}}")
  else:
    assert current_player == -1

  legal_actions = get_legal_actions(state)
  chosen_action = rg.choice(legal_actions)
  print(f"Taking action {{repr(chosen_action)}} from {{len(legal_actions)}} options,
      first 10 are {{[*legal_actions][:10]}}")
  state = apply_action(state, chosen_action)
else:
  raise ValueError(f"Game did not end after 1000 steps.")
```

This tests that if every player randomly picks a valid move, the game will correctly play and terminate. Note that we assume access to the static and deterministic initial state of the game, before any chance nodes have taken place. This could also be synthesized by the LLM instead.

### H.6 RESAMPLING THE STATE AT GAME PLAYING TIME FOR IMPERFECT INFORMATION GAMES

When playing the game, we allow the system to up to 10 tries to get a valid state that produces the current observations:

```
for retry in range(10):
  json_state = {start_state}
  try:
    actions = resample_history(obs_action_history, player_id)
    for action in actions:
      json_state = apply_action(json_state, action)
      state_log.append(json_state)
  except Exception as e: # Running generated code, could raise anything.
    continue
  recreated_obs = get_observations(json_state)[player_id]
  if recreated_obs == obs_action_history[-1][0]:
    return json_state
```

Additionally, if the ISMCTS process fails due to, e.g., poor understanding of the game termination criteria in the CWM, we fall back to resampling the state and then return a uniformly sampled legal action from that state.

### H.7 VALUE FUNCTION SYNTHESIS

Our value function synthesis function prompt is

```
'''
You are an expert python programmer. You are playing the game {game}, and need
to synthesize a value function for monte carlo tree search.

{game_description}

For reference, the game is implemented as follow

{code}

The function you need to write is:
{value_function}

It should return the reward at terminal states, and otherwise an estimate of the
value for each non-terminal states.

It should always be a float:
{player_tests}

Terminal states should match rewards:
{terminal_tests}

To write a good value function first reason about the game and produce a heuristic value
      that is informative, and do not just output zeros everywhere other than terminal
    states.
Finally ONLY output the new value_function, do not output any other text, code,
explanations or placeholders.
The response code must be a single CODE BLOCK that uses this format:
The opening fence: ```python
The closing fence: ```
'''
```

Where `value_function` is

```
'''
def value_function(state: dict[str, Any], player_id: int) -> float:
  """Returns the value estimate for player_id in state.
```

```
   For terminal states the function returns the true return. For ongoing play
   the function should return a value estimate that reflect the winning potential
   of the player with given player_id.
   """
'''
```

`player_tests` and `terminal_tests` is the list of example

```
'''
{current_player}
self.assertIsInstance(value_function(state, {current_player}), float)
if {current_player} == pyspiel.PlayerId.TERMINAL:
    rewards = get_rewards(state)
    for player in range(len(rewards)):
        self.assertEqual(rewards[player], value_function(state, player))
'''
```

# I GAME RULES

## I.1 BACKGAMMON

Backgammon is a two-player board game that combines strategy and luck. The object is to
    move all of your checkers off the board before your opponent does. Here's a
    breakdown of the rules:

**The Board and Setup:**

* **The Board:** The board consists of 24 narrow triangles called **points**. These
    points are grouped into four quadrants of six points each:
    * **Inner/Home Board:** The quadrant closest to each player's starting position.
    * **Outer Board:** The quadrant further from each player's starting position.
* **The Bar:** The area in the middle of the board, separating the two sides.
* **The Bear-Off Area:** The area off the board where checkers are moved once they reach
      the player's home board.
* **Checkers:** Each player has 15 checkers of one color (typically black and white).
* **Dice:** Two dice are used to determine movement.
* **Doubling Cube (Optional but common):** A cube with the numbers 2, 4, 8, 16, 32, and
    64, used to increase the stakes of the game.

**Initial Setup:**

Each player's 15 checkers are set up in a specific configuration on the points:

* 2 checkers on the opponent's 24-point.
* 5 checkers on the opponent's 13-point.
* 3 checkers on their own 8-point.
* 5 checkers on their own 6-point.

**Gameplay:**

1. **Starting the Game:** Each player rolls one die. The player with the higher roll
    goes first. If the rolls are the same, they roll again until one player rolls
    higher. The player who goes first uses the numbers rolled on *both* dice to make
    their first move.

2. **Rolling the Dice:** On subsequent turns, each player rolls two dice.

3. **Moving Checkers:** After rolling the dice, the player must move their checkers
    according to the numbers rolled.
    * **Separated Moves:** Each die represents a separate move. You can move one checker
        the distance of one die's roll and another checker the distance of the other
        die's roll.
    * **Combined Move:** You can move one checker the combined distance of both dice
        rolls, but *only if* the point you would land on for the first die's roll is not
        blocked (see "Blocked Points" below).
    * **Mandatory Moves:** You must move your checkers if possible. If you can only make
        one of the two moves indicated by the dice, you must make that move. If you can
        make both, you must make both.
    * **No Legal Moves:** If you cannot make any legal moves based on the dice roll,
        your turn ends.

4. **Point Direction:** You always move your checkers from your opponent's inner board
    towards your own home board. The points are numbered 1 to 24, where 24 is the
    latest point in your opponent's inner board. Each move makes the checker move to
    smaller numbered points.

5. **Landing on a Point:**
    * **Empty Point:** You can land on an empty point.
    * **Point Occupied by Your Own Checkers:** You can land on a point occupied by any
        number of your own checkers.
    * **Point Occupied by Opponent's Checkers:**

* **Blots:** If a point is occupied by *only one* of your opponent's checkers, it 's called a "blot." If you land on a blot, you "hit" the opponent's checker. The hit checker is placed on the **bar**.
* **Blocked Points:** If a point is occupied by *two or more* of your opponent's checkers, it is "blocked." You *cannot* land on a blocked point.

6. **Entering from the Bar:** If a player has checkers on the bar, they must re-enter them onto the board before making any other moves.
   * **Re-entry Points:** You can re-enter a checker from the bar onto a point in your opponent's home board that corresponds to the number rolled on a die. For example, if you roll a 3, you can re-enter a checker onto your opponent's 3-point.
   * **Blocked Re-entry:** If the corresponding point in your opponent's home board is blocked by two or more of your opponent's checkers, you cannot re-enter using that die roll.
   * **Priority:** You must use any available die rolls to re-enter checkers from the bar. If you can re-enter one checker but not the other based on your dice roll, you must still re-enter the one you can. If you cannot re-enter any checkers, your turn ends.

7. **Doubles:** If you roll doubles (e.g., two 4s), you can use each number *four times *. So, two 4s means you have four moves of 4. You can use these moves in any combination, as long as they are legal. Each turn allows you to make two moves only . So if a player rolls a double, they take an extra turn, making at most two moves in each of the turns.

8. **Bearing Off:** If you don't have any checkers outside of your home board or at the bar, you can "bear off" chekers (moving them off the board) that are at your home board.
   * **Bearing Off Rolls:** To bear off a checker, you must roll the exact number that the checker is on to move it off the board from its current point. For example, if a checker is on your 4-point, you need to roll a 4 to bear it off.
   * **Higher Rolls:** If you roll a number higher than the highest point occupied by your pieces, you can still bear off a piece. However, you must bear off a piece from the highest occupied point. For example, if your highest occupied point is the 4-point, and you roll a 6, you can bear off a piece only from the 4-point, you cannot bear off a piece from lower points.
   * **Lower Rolls:** If you roll a number lower than the point your checker is on, you can still move a checker from a higher point the distance of the roll (if legal ), or you must move a checker from a lower point the distance of the roll if possible. You cannot bear off a checker if you have checkers on higher points in your home board that can be moved by the dice roll.
   * **Blocked Bear Off:** You cannot bear off a checker if any of your checkers are still on the bar or outside of your home board. You must bring all your checkers into your home board before bearing off.
   * **Moving Pieces Within Home Board:** Instead of bearing off, you can also move your checkers within your home board using the dice rolls.

9. **Action Notation:** Player moves are typically represented using a specific notation . Each turn consists of at most two moves. Each move isrepresented by a string of the form "Move checker at X using Y roll", where "X" is the position of the checker being moved, and "Y" indicates if the move is done based on the dice roll with the higher or lower number.
   The first position is always a number between 1 and 24 or "Bar", and it is presented in each player's perspective, where 24 is the latest point in the opponent's inner board.
   For a double roll (e.g., a 3-3, granting four moves of 3 as per rule 7 "Doubles") :
     - It is assumed these four moves are made by the player in two stages: First, providing two moves, then the player gets a second turn then providing the subsequent two moves. For instance, a player might move two checkers, each making two 3-point moves. Next, the player would have a second turn, providing two more moves of 3.

10. **Board Notation:** The board is represented with 2 arrays of 24 numbers,

    where each number is either 0 (empty) or a number between 1 and 15
    (indicating the number of checkers of that color on that point).
    The first array is for the first player, and the second is for the second.
    The board ordering is from second player's persective. Starts from first
    player's home base and ends at the second player's home base. The first
    index is the latest point in the second player's inner board, meaning
    position 24 for the first player and 1 for the second player.

**Winning the Game:**

The first player to bear off all 15 of their checkers wins the game.

**Optional Rules (Commonly Used):**

* **The Doubling Cube:**
    * **Offering a Double:** At the start of their turn, *before* rolling the dice, a
        player can offer to "double" the stakes of the game.
    * **Accepting a Double:** The opponent can either accept or decline the double. If
        they decline, they lose the game immediately and the current stake is paid. If
        they accept, the stakes are doubled, and the opponent now "owns" the doubling
        cube, meaning they are the only one who can offer the next double.
    * **Subsequent Doubles:** The owner of the cube can offer to redouble at the start
        of their turn. The stakes continue to double with each accepted redouble (2, 4,
        8, 16, etc.).
* **Gammon and Backgammon:** These are ways to win with higher stakes.
    * **Gammon:** If a player bears off all their checkers before the opponent has borne
        off *any* checkers, the winner wins a "gammon," which is typically worth double
        the value of the doubling cube.
    * **Backgammon:** If a player bears off all their checkers before the opponent has
        borne off *any* checkers and the opponent still has one or more checkers on the
        bar or in the winner's home board, the winner wins a "backgammon," which is
        typically worth triple the value of the doubling cube.

**Key Concepts and Strategy:**

* **Hitting Blots:** Hitting your opponent's checkers puts them on the bar and disrupts
    their progress.
* **Making Points:** Occupying points with two or more of your checkers creates "blocks"
    that prevent your opponent from moving past. Strategic point-making is crucial.
* **Prime:** Creating a "prime" (six consecutive blocked points) can severely hinder
    your opponent's movement.
* **Running:** Moving your checkers quickly towards your home board.
* **Positioning:** Carefully considering where to move your checkers to maximize your
    options and limit your opponent's.
* **Risk vs. Reward:** Balancing the risk of leaving blots with the potential for making
    good moves.

Backgammon is a game with layers of strategy that unfold as you play. While the dice
    introduce an element of chance, skillful play, understanding probability, and
    strategic decision-making significantly influence the outcome. Enjoy the game!

## I.2 CONNECT FOUR

### Rules of Connect Four

*   **Setup:** Connect Four is played on a 6-row by 7-column vertical grid, which starts
     completely empty.
*   **Players and Marks:** There are two players: Player 0 uses the 'x' mark and Player
     1 uses the 'o' mark.
*   **Turns:** Player 0 ('x') always goes first, and turns alternate between players.
*   **Making a Move:** On your turn, you choose a column to drop your mark into. The
     mark will fall to the lowest unoccupied square within that chosen column.
     Attempting to drop a mark into a column that is already full is an invalid move;
     you must choose a column with at least one empty square to complete your turn.

* **Winning the Game:** The winner is the first player to get four of their marks in a
  row (horizontally, vertically, or diagonally). The game ends immediately as soon
  as a winning line is formed.
* **Drawing the Game:** If all 42 squares on the grid are filled and neither player
  has won, the game ends in a draw.
* **End the Game:** The game only concludes upon a win or a draw. A player must make a
  move on their turn as long as there is at least one valid move available on the
  board.
* **Move Notation:** Use the move notation '[mark][col]', where col is the 0-indexed
  column you are dropping your mark into. For example, 'x3' means Player 0 ('x')
  drops their mark into the fourth column from the left (column index 3).

## I.3   TIC-TAC-TOE

### Rules of Tic-Tac-Toe

* **Setup:** Tic-Tac-Toe is played on a 3x3 grid, which starts completely empty.
* **Players and Marks:** There are two players: Player 0 uses the 'x' mark and Player
  1 uses the 'o' mark.
* **Turns:** Player 0 ('x') always goes first, and turns alternate between players.
* **Making a Move:** On your turn, you must place your mark in a single, unoccupied
  square. Attempting to place a mark in an already occupied square is an invalid move
  ; you must choose an empty square to complete your turn.
* **Winning the Game:** The winner is the first player to get three of their marks in
  a row (horizontally, vertically, or diagonally). The game ends immediately as soon
  as a winning line is formed.
* **Drawing the Game:** If all nine squares on the grid are filled and neither player
  has won, the game ends in a draw.
* **End the Game:** The game only concludes upon a win or a draw. A player must make a
  move on their turn as long as there is at least one valid move available on the
  board.
* **Move Notation:** Use the move notation 'mark(row,col)', where row and col are 0-
  indexed. For example, 'x(0,0)' means Player 0 ('x') places their mark in the top-
  left square.

## I.4   GEN. TIC-TAC-TOE

Generalized Tic-Tac-Toe (6x6, Win Length 4, 2 Players)

1. Overview: This is a two-player strategy game played on a 6x6 grid. The goal
is to be the first player to form a continuous line of four of your own marks.
This game is a specific configuration of a generalized Tic-Tac-Toe framework.

2. Game Setup:

Board: A 6x6 grid of cells (36 cells in total), with rows and columns numbered
0 to 5.
Players: Two players. Conventionally, one player uses 'x' and the other uses 'o'.
Starting State: The board is initially empty.

3. Gameplay:

Players take turns placing their mark on an unoccupied cell on the board.
A designated player (e.g., Player 'x') makes the first move.
The game continues with players alternating turns.

4. Winning Condition:

A player wins if they are the first to place four of their marks in an unbroken
straight line.

This line can be:

```
* Horizontal: Four marks in the same row.
* Vertical: Four marks in the same column.
* Diagonal: Four marks along any of the board's diagonal lines (both directions).

5. Draw Condition:

If all cells on the 6x6 board are filled with marks, and neither player has
achieved a line of four of their marks, the game is a draw.

6. End of Game:

The game concludes immediately when either:
One player achieves a winning line of four marks (that player is the winner).
All cells are filled, and no winning line exists (the game is a draw).

7. Key Parameters for this Specific Variant:

Number of Rows: 6
Number of Columns: 6
Winning Line Length: 4
Number of Players: 2
```

## I.5   GEN. CHESS

```
The game of generalized chess is a two player game where each player controls a
collection of pieces and wins by capturing the target piece from the other
player. Each kind of game piece has a specific pattern of movements that it can
execute. A piece can execute any one of its available moves as long as that move
stays on the board and doesn't land on another of that player's pieces. If the
piece lands on an opponent piece, it captures the opponent piece and removes it
from the board. Allowed piece movements are not the same as in standard chess.

Actions are described using board coordinates. For a 5x5 board, rows are labeled
A-E from top to bottom, and columns are labeled 1-5 from left to right. A move
from a starting square to a destination square is written as 'start_to_end',
for example, 'A2_to_C2' means move the piece from square A2 to square C2.

Passing a turn is specified as 'PASS'.

This 'army5x5a' variant of generalized chess is played on a 5x5 board.

It includes the following pieces, with their corresponding set of allowed moves:
 - general: [(1, 0), (-1, 0), (0, 1), (0, -1), (0, -2), (0, 2)]
 - infantry: [(1, 0), (2, 0), (1, -1), (1, 1), (-1, 0)]
 - cavalry: [(0, 3), (1, 2), (2, 1), (3, 0)]

Game pieces are depicted with the following symbols: 'general': 'X', 'infantry': 'I', '
    cavalry': 'V'. Player 0 pieces are upper-case while Player 1 pieces are lower-case.

The 'general' is the target piece. Capturing this piece wins the game.
```

## I.6   BARGAINING

```
The rules of "bargaining" aren't fixed and formal like a board game with a rulebook.
    Instead, it's a dynamic social process of negotiation where two or more parties
    attempt to reach a mutually agreeable outcome on a price or terms for a product,
    service, or agreement. Here's a breakdown of the core principles and common "rules"
     of bargaining, understood more as strategies and expectations:

**Core Principles of Bargaining:**
```

* **Mutual Desire for an Agreement:** Both parties generally want to reach a deal, even if their initial positions are far apart.
* **Information Asymmetry:** One party often has more information than the other, which can influence the negotiation.
* **Iterative Process:** Bargaining usually involves a series of offers and counter-offers.
* **Focus on Value:** Bargaining is about perceived value – what each party believes the item or service is worth.
* **Potential for Compromise:** Both parties are usually expected to give a little to reach an agreement.

**Implicit "Rules" or Common Strategies:**

These are not hard-and-fast rules, but rather common practices and expectations that guide the negotiation:

1. **Know Your Limits (Walk-Away Point):** Before starting, each party should have a clear idea of the maximum (for a buyer) or minimum (for a seller) price they are willing to accept. This is your "reservation point."

2. **Start with an Anchor (Opening Offer):** The first offer sets an "anchor" for the negotiation. This is usually a price lower than what the seller expects (for a buyer) or higher than what the buyer expects (for a seller).
   * **Seller's Perspective:** Start higher than your desired price.
   * **Buyer's Perspective:** Start lower than what you're willing to pay.

3. **Justify Your Offers:** Simply stating a price is less effective than explaining *why* you're offering that price. Reference market value, condition of the item, your budget, etc.

4. **Make Concessions Incrementally:** Don't jump straight to your walk-away point. Make small concessions with each counter-offer. This signals a willingness to negotiate while still trying to get the best possible deal.

5. **Signal Willingness to Walk Away (But Don't Bluff Too Much):** Letting the other party know you're willing to walk away if you don't get a satisfactory price can be a powerful tactic. However, repeated or unbelievable threats can undermine your credibility.

6. **Listen Actively and Ask Questions:** Pay attention to the other party's offers, reasoning, and potential underlying needs. Asking questions can reveal information and build rapport.

7. **Be Patient:** Bargaining takes time. Don't rush the process.

8. **Maintain a Respectful Tone:** Even if the negotiation becomes difficult, try to maintain a polite and respectful demeanor. Aggression can shut down the conversation.

9. **Consider Non-Price Factors:** While price is central, bargaining can also involve other terms like delivery time, payment method, warranties, or additional items included.

10. **Know When to Stop:** If it's clear you won't reach an agreement that meets your needs, it's okay to respectfully end the negotiation.

11. **Be Prepared to Walk Away:** If you can't reach an agreement within your limits, you must be prepared to walk away. This is crucial for maintaining your boundaries.

12. **The Final Offer:** Often, one party will indicate their "final offer." This suggests they are unwilling to make further concessions. However, this isn't always truly final and can be tested with a counter-offer.

13. **The Art of the Counter-Offer:** Respond to offers with a counter-offer that is a concession from your previous position, but still moves you closer to your goal.

**Situational Differences:**

The "rules" of bargaining can vary depending on the context:

* **Cultural Norms:** Bargaining is much more common and expected in some cultures (e.g., bazaars in many parts of the world) than others (e.g., retail stores in most Western countries).
* **Type of Item/Service:** Bargaining for a car is different than bargaining for a small trinket at a market.
* **Power Dynamics:** Who has more leverage in the negotiation can significantly impact the process.

**In summary, the "rules" of bargaining are less about strict regulations and more about strategic communication, understanding the other party's perspective, and being prepared to make concessions to reach a mutually acceptable agreement. It's a negotiation dance where both parties are trying to get the best possible outcome within their own limits.**

## I.7 LEDUC POKER

Leduc Poker is a simplified two-player poker game, ideal for AI research, that uses a small deck to focus on core poker concepts like betting strategy and imperfect information.

Here is a detailed breakdown of the rules to clarify legal moves. Note that in this implementation, the "Check" action is not available; players must use "Call" instead. A call may be zero-cost if there is no outstanding bet to match.

**1. Setup & Preliminaries**
*   **Players:** 2.
*   **Deck:** 6 cards (two Jacks, two Queens, two Kings).
*   **Blinds:** Before cards are dealt, mandatory bets are posted:
    *   Player 1 (P1) posts a **Small Blind** of 1 unit.
    *   Player 2 (P2) posts a **Big Blind** of 2 units.
*   **The Deal:** Each player receives one private card, face down.

**2. Core Betting Rules**
*   **Raise Sizing:** The amount to raise is fixed.
    *   **Round 1:** The raise amount is **2 units**.
    *   **Round 2:** The raise amount is **4 units**.
*   **Total Betting Cap:** The total betting cap for each round is a maximum of **two raises**.
*   **Acting First:** Player 1 (the small blind) acts first in both betting rounds (pre-flop and post-flop).

**3. Round 1: Pre-Flop Betting**
This round occurs before the public card is revealed.

*   **P1's First Action:** P1 must act on P2's 2-unit Big Blind.
    *   **Fold:** Forfeit the 1-unit blind. P2 wins the pot.
    *   **Call:** Match the 2 units by putting in 1 more unit.
    *   **Raise:** Make a 2-unit raise, for a total of 4 units (P1 puts in 3 units). The total betting cap has been reached.
*   **P2's Action:**
    *   If P1 **called**, P2 can **Call** (a zero-cost action, as bets are equal) to end the round, or **Raise** (by putting in 2 more units to make it 4 total).
    *   If P1 **raised**, P2 can only **Call** (by putting in 2 more units) or **Fold**. The betting cap has been reached.
*   **P1's Second Action (if necessary):** If P1 called and P2 then raised, the action returns to P1. P1 can only **Call** (by putting in 2 more units) or **Fold**.

**4. The Flop: Public Card**

After Round 1 betting concludes, one public card is dealt face-up. This card is shared
    by both players.

**5. Round 2: Post-Flop Betting**
This round occurs after the flop. There are no blinds.

*   **P1's First Action:**
    *   **Call:** Make a zero-cost call to pass the turn (as there is no outstanding bet)
        .
    *   **Raise:** Make a 4-unit raise.
*   **P2's Action:**
    *   If P1 **called** (at zero-cost), P2 can also **Call** (at zero-cost, ending the
        round) or **Raise** 4 units.
    *   If P1 **raised**, P2 can **Call** (matching the 4 units), **Raise** (by putting
        in another 4 units, for a total bet of 8), or **Fold**. The total betting cap
        has been reached.
*   **Subsequent Actions:**
    *   If P2 **raised** (after P1's initial zero-cost call), the action returns to P1,
        who can **Call** (the 4 unit bet), **Raise** (to 8 total), or **Fold**. The
        total betting cap has been reached.
    *   If a player **raises**, the other player can only **Call** or **Fold**, as the
        betting cap has been reached.

**6. Showdown & Hand Ranking**
If neither player folds, a showdown occurs after Round 2 betting.

*   **Hand:** A player's hand is their private card combined with the public card.
*   **Hand Ranks (best to worst):**
    1.  **Pair:** Two cards of the same rank (e.g., J-J). Higher pairs beat lower pairs.
    2.  **High Card:** If no one has a pair, the player with the highest card wins (K > Q
        > J).
*   **Ties:** If both players have the same hand rank (e.g., both have a King-high), the
    pot is split.

**7. Winning**
A player wins the pot either by being the only one left after the other folds, or by
    having the best hand at showdown.

## I.8   GIN RUMMY

# The Game of Gin Rummy

Gin Rummy is a two-player card game played with a standard 52-card deck. The
primary objective is to form "melds" in your hand, which are either sets of
three or four cards of the same rank (e.g., 7h 7c 7d) or runs of three or more
cards of the same suit in sequence (e.g., 4h 5h 6h). Cards not part of any meld
are referred to as "deadwood." The value of deadwood cards corresponds to their
rank (Aces are 1 point, face cards are 10, and number cards are their face
value). The ultimate goal is to minimize the point value of your deadwood.

A round of Gin Rummy concludes when a player "knocks." A player can choose to
knock on their turn if the total point value of their deadwood is less than or
equal to a predetermined "knock card" value. Announcing "gin" is a special type
of knock where a player has no deadwood at all.

# Player Hand Information: This section provides details about your own hand.

Deadwood: This calculates the current point total of the cards in
your hand that are not part of a valid meld (a set or a run). Minimizing this
value is the primary goal.

The Card Grid: This is a visual representation of the cards you currently hold.
It is organized logically for easy parsing:

Rows: Each of the four rows corresponds to a suit, in the order of Spades (top row), Clubs, Diamonds, and Hearts (bottom row).

Columns: The columns represent the rank of the cards, ordered from Ace on the far left to King on the far right.

Here are also some example moves:

Player: 0 Action: Pass
Player: 1 Action: Draw upcard
Player: 1 Action: Jc
Player: 0 Action: 3d
Player: 1 Action: Draw stock

# Action Legality is Dictated by Game Phase: Before selecting a move, you must first check the phase.

If the phase is Draw, the only valid actions are Draw upcard or Draw stock.

If the phase is Discard, the only valid actions are to discard a specific card from your hand (e.g., Action: 4c) or to Knock.

A player cannot discard a card until after they have successfully drawn one.

# Special Case: The First Turn of the Round

The very first turn of a round has a unique rule. The non-dealer has the first option on the initial upcard.

The non-dealer can either take the upcard (Draw upcard) or Pass.

If the non-dealer passes, the dealer then has the same choice: take the upcard or pass.

If both players pass on the initial upcard, the non-dealer must then start their turn by drawing from the stock pile. After this initial sequence, play continues with the standard draw/discard phases.

# Knocking:
When a player knocks in Gin Rummy, the round immediately ends and a specific sequence of scoring, known as the "layoff," begins. Here is a detailed breakdown of what happens.

1. The Knock and Laying Down Hands
First, the player who is knocking (the "knocker") lays their hand face up on the table, organising their cards into melds (sets and runs) and separating their unmelded cards, known as "deadwood."

What the Player Needs to Do After Knocking
After sending Action: Knock, the player must follow a strict, multi-step process to lay down their hand for scoring.

Step 1: Declare Your Melds
The player must now explicitly declare their melds to the game, one by one.
For exampple, if the agent's hand contains two valid runs:

A run of clubs: 7c8c9cTc

A run of diamonds: 9dTdJdQd

Correct First Move in the Knock Phase:

```
Player: 1 Action: 7c8c9cTc
or
Player: 1 Action: 9dTdJdQd
```

Step 2: Declare Subsequent Melds
After the agent declares its first meld, it will receive a new observation. The game will still be in Phase: Knock. The Valid actions will now include any remaining melds that can be made from the cards left in the hand.

Note: The value of the knocker's deadwood must be 10 points or less (or the value of the designated knock card for that round). Face cards are worth 10 points, aces are 1 point, and all other cards are their numerical value.

2. The Opponent's Turn: Laying Off
Next, the defending opponent lays down their own hand, also separating their melds from their deadwood. Crucially, the opponent then gets the opportunity to "lay off" any of their own deadwood cards by adding them to the knocker's melds.

For example:

If the knocker has a meld of three Kings (Ks Ks Ks), and the opponent has the fourth King (Ks) as deadwood, they can add it to the knocker's set, thus eliminating those 10 points from their deadwood count.

If the knocker has a run of 5h 6h 7h, the opponent can lay off a 4h or an 8h to extend the run.

The knocker is not allowed to lay off any of their deadwood on the opponent's melds.

3. Scoring the Hand
After the opponent has finished laying off their cards, both players calculate the final value of their remaining deadwood. The scoring for the hand is then determined in one of three ways:

a) A Successful Knock
If the knocker's deadwood count is lower than the opponent's deadwood count, the knocker scores the difference between the two counts.

Example: The knocker has 7 points of deadwood. The opponent initially has 35 points, but after laying off a 10-point card, their deadwood is reduced to 25. The knocker scores 18 points (25 - 7).

b) An Undercut
If the opponent, after laying off their cards, has a deadwood count that is equal to or less than the knocker's count, they have "undercut" the knocker. In this scenario, the opponent scores the difference in points (if any) plus a bonus, which is typically 25 points.

Example: The knocker has 8 points. The opponent has 6 points after layoffs. The opponent scores 2 points (8 - 6) plus a 25-point bonus, for a total of 27 points.

c) Going Gin
If the knocker has a deadwood count of zero, this is called "going gin." The knocker receives a bonus (typically 25 points) in addition to the full value of the opponent's entire deadwood count. When a player goes gin, the opponent is not allowed to lay off any of their cards.

Example: A player goes gin. Their opponent has 42 points of deadwood. The ginning player scores 42 points plus a 25-point gin bonus, for a total of 67 points.

What if the Stock Pile Runs Out?
If the stock pile is reduced to its last two cards and the player who drew the

third to last card discards without knocking, the hand is declared a draw. No points are awarded to either player, and the deal passes to the next player for a new round.

# If you see Phase: Wall in an observation, it means:

The Stock Pile is Exhausted: The round has concluded because there are no more cards to be drawn from the stock.

No Player Has Knocked: Neither you nor your opponent were able to knock by the time the last card was drawn.

The Hand is a Draw: No points are awarded to either player for this round. The hand is over.

No Action is Required: The game is in a terminal state for the current round. The only thing to do is to acknowledge the result and wait for the next hand to be dealt. The deal will typically pass to the player who didn't deal the drawn hand. Thus `Player: X Action: Pass` must be provided as action.

## I.9 QUADRANTO

Quadranto is a partially observable game in which two players try to catch each other in a 4 by 4 matrix.

The 4 by 4 matrix is divided in 4 quadrants. At the beginning, player 0 is randomly placed in the top left quadrant and player 1 is randomly placed in the bottom right quadrant.

During their turn, each player can choose to move in each of the four cardinal directions, "Left", "Right", "Up", "Down". Or they can choose to "Stay", which means they remain where they are. When a player moves, if it lands on the same location where the other player is, it wins and the game ends.

The observation tells the player where it is located and in which *quadrant* the opponent player is located. Therefore, neither player knows exactly where the other player is located until the very moment in which one player catches the other.

If the players perform a total of 20 moves without catching each other, the game ends in a draw, both players get 0 points. If one catches the other, the winning player gets +1 points and the losing player gets -1 points.

## I.10 HAND OF WAR

**Hand of War** is a strategic card game where choosing your cards wisely is key to victory. You'll manage a hand of cards, adding a layer of tactical decision-making to every round as you aim to capture all of your opponent's cards.

**Objective:**

*   The goal of Hand of War is to capture as many of your opponent's cards.

**Setup:**

*   **Shuffle and Deal:** Thoroughly shuffle the deck. Deal the entire deck evenly between two players, face down.
*   **Form Hands:** Each player draws the top three cards from their draw pile.

**Gameplay (The "Battle"):**

*   **Choose a Card:** Simultaneously, both players select one card from their hand and place it face down.
*   **Reveal and Compare:** Both players flip their chosen cards.
*   **Higher Card Wins:** The player with the higher-ranking card wins the battle and takes both cards, placing them at the bottom of their win pile.
*   **Card Ranking:** Ace (High), K, Q, J.
*   **Draw New Cards:** After the battle, players draw from their draw pile to replenish their hand to three cards.

**"Showdown" (When Cards Tie):**

*   **Declaration:** If cards are of the same rank, a "Showdown" occurs.
*   **Face-Down Cards:** Each player places 1 card from their draw pile face down.
*   **Choose Battle Card:** Players choose one card from their hand and place it face up.
*   **Determine Showdown Winner:** Higher battle card wins all cards in the Showdown.
*   **Another Tie:** Repeat Showdown process (burn 1, choose card).
*   **Draw After Showdown:** Players replenish their hand to three cards.

**Game End Conditions:**

The game can end in one of two ways:

1.  **Winning by Capturing All Cards:** The game ends immediately if one player possesses all 16 cards. That player is the winner.

2.  **Winning by Win Pile Count (Draw Pile Depleted):** If a player's draw pile becomes empty and they are required to perform an action they cannot complete (such as drawing a card to replenish their hand after a battle, or burning a card during a Showdown), the game ends immediately. In this scenario:
    *   The number of cards in each player's win pile is counted.
    *   The player with more cards in their win pile wins the game.
    *   If both players have an equal number of cards in their win piles, the game results in a draw.
    *   Note: The win pile is never reshuffled into the draw pile during play.

**Notation**
Player 0 always moves first, followed by player 1. Thus if the previous move was a player 0 move, then the next move is a player 1 move.

For convenience, a list of played cards is provided in PubliclyRevealedCards, but not playable anymore.

## J  SAMPLE SYNTHESIZED CWMS AND INFERENCE FUNCTIONS

We show the resulting CWM after tree search refinement on perfect and imperfect information games, and for the latter we include both open and closed deck inference. This includes inference and value functions when they exist.

### J.1  TIC-TAC-TOE (PERFECT INFORMATION)

```python
import collections
from typing import Any, cast

# Type aliases for clarity as defined in the problem.
Action = str
State = dict[str, Any]
PlayerObservation = dict[str, Any]

# Game constants for mapping players, marks, and defining winning conditions.
_MARK_TO_PLAYER = {'x': 0, 'o': 1}
_WINNING_LINES = [
    # Rows
    (0, 1, 2), (3, 4, 5), (6, 7, 8),
    # Columns
    (0, 3, 6), (1, 4, 7), (2, 5, 8),
    # Diagonals
    (0, 4, 8), (2, 4, 6)
]

# --- Helper Functions ---

def _check_winner(board: list[str | None]) -> str | None:
    """Checks if there is a winner on the board, returning the winner's mark."""
    for line in _WINNING_LINES:
        p1, p2, p3 = line
        # Check if all three cells in a line are the same and not empty.
        if board[p1] and board[p1] == board[p2] == board[p3]:
            return board[p1]
    return None

def _is_game_over(board: list[str | None]) -> bool:
    """Checks if the game has ended either by a win or a draw."""
    return _check_winner(board) is not None or all(cell is not None for cell in board)

# --- Core Game Functions ---

def apply_action(state: State, action: Action) -> State:
    """Returns the new state after an action has been taken."""
    mark = action[0]
    row = int(action[2])
    col = int(action[4])

    # Create a copy of the board to modify.
    new_board = state['board'][:]
    index = row * 3 + col
    new_board[index] = mark

    # A game is over if there is a winner or the board is full.
    if _is_game_over(new_board):
        next_player_mark = None
    else:
        # Alternate turns between 'x' and 'o'.
        next_player_mark = 'o' if mark == 'x' else 'x'

    return {'board': new_board, 'current_player_mark': next_player_mark}

def get_current_player(state: State) -> int:
    """Returns current player, with -1 for chance and -4 for terminal."""
    mark = state['current_player_mark']
    if mark is None:
        return -4  # Terminal state
    return _MARK_TO_PLAYER[mark]

def get_player_name(player_id: int) -> str:
    """Returns the name of the player, with 'chance' for -1, and 'terminal' for -4."""
    if player_id == -4:
        return 'terminal'
    if player_id == -1:
        return 'chance'
    return str(player_id)

def get_rewards(state: State) -> list[float]:
    """Returns the rewards per player from their last action."""
    # Rewards are only non-zero in a terminal state.
    if get_current_player(state) != -4:
        return [0.0, 0.0]

    winner_mark = _check_winner(state['board'])
    if winner_mark == 'x':
```

```python
    return [1.0, -1.0]  # Player 0 wins
  if winner_mark == 'o':
    return [-1.0, 1.0]  # Player 1 wins

  return [0.0, 0.0]  # Draw

def get_legal_actions(state: State) -> list[Action]:
  """Returns legal actions that can be taken in current state."""
  # No legal actions if the game is over.
  if get_current_player(state) == -4:
    return []

  actions = []
  mark = state['current_player_mark']
  # An action is legal for any empty (None) square.
  for i, cell in enumerate(state['board']):
    if cell is None:
      row, col = divmod(i, 3)
      actions.append(f"{mark}({row},{col})")
  return actions

def get_observations(state: State) -> list[PlayerObservation]:
  """Returns the observation for player."""
  # Fully observable game, so both players see the complete state.
  return [state, state]
```

## J.2 Bargaining (imperfect information, open deck)

```python
import copy
import itertools
import random
from typing import Any, Dict, List, Optional, Tuple

# Type Aliases for clarity
State = Dict[str, Any]
Action = str
PlayerObservation = Dict[str, Any]

# --- Game Constants ---
ITEMS = sorted(['X', 'Y', 'Z'])
NUM_PLAYERS = 2
MAX_TURNS = 10
MAX_ITEM_VALUE = 10
# This is a fixed set of chance outcomes, likely for deterministic testing or analysis.
```

```python
_CHANCE_ACTIONS = sorted(['1,2,3;8,1,0;4,0,2',  '1,4,1;4,1,2;2,2,0',  '2,2,1;1,1,6;0,4,2',  '1,4,1;9,0,1;2,2,0',  '1,4,1;5,1,1;0,1,6',
'4,1,1;2,1,1;1,0,6',  '3,1,1;1,4,3;0,2,8',  '1,1,3;0,1,3;1,3,2',  '1,3,1;2,2,2;10,0,0',  '1,2,2;2,3,1;4,0,3',  '1,4,1;6,1,0;8,0,2',
'1,1,3;7,3,0;0,4,2',  '1,5,1;4,0,6;3,1,2',  '3,3,1;3,0,1;0,2,4',  '1,2,3;8,1,0;7,0,1',  '4,1,2;0,6,2;2,2,0',  '2,1,2;3,2,1;4,2,0',
'1,3,1;4,2,0;8,0,2',  '2,1,3;3,1,1;0,10,0',  '1,3,1;6,1,1;4,1,3',  '2,2,1;3,0,4;2,1,4',  '3,3,1;1,1,4;3,0,1',  '1,2,3;0,5,0;3,2,1',
'1,3,1;1,2,3;3,1,4',  '4,1,1;0,0,10;1,3,3',  '2,4,1;2,1,2;2,1,2',  '4,1,2;1,6,0;1,2,2',  '1,1,4;4,2,1;4,6,0',  '1,5,1;2,0,8;5,1,0',
'1,3,1;0,1,7;6,0,4',  '1,1,4;4,6,0;0,2,2',  '1,1,5;3,2,1;2,8,0',  '1,3,2;7,1,0;4,0,3',  '2,1,3;1,2,2;2,3,1',  '1,3,1;0,1,7;7,0,3',
'1,3,1;2,2,2;1,2,3',  '1,5,1;9,0,1;0,1,5',  '4,1,1;0,4,6;1,5,1',  '2,2,1;0,2,6;4,1,0',  '3,1,1;2,1,3;0,6,4',  '1,1,3;10,0,0;1,3,2',
'3,2,1;2,1,2;1,3,1',  '1,3,1;5,1,2;3,0,7',  '1,4,1;1,2,1;3,0,1',  '4,2,1;1,3,0;0,3,4',  '2,2,1;1,3,2;5,0,0',  '1,3,1;4,2,0;1,1,6',
'1,1,3;6,1,1;0,1,3',  '2,1,2;3,4,0;3,2,1',  '1,4,1;2,1,4;9,0,1',  '2,2,2;0,3,2;1,3,1',  '3,3,1;0,2,4;1,0,7',  '3,1,1;1,0,7;0,8,2',
'4,1,1;1,4,2;2,1,1',  '1,3,1;0,0,10;1,1,6',  '2,2,1;3,0,4;2,3,0',  '2,2,2;2,3,0;0,4,1',  '2,1,2;3,4,0;1,2,3',  '3,1,1;2,2,2;0,2,8',
'1,2,2;4,0,3;2,1,3',  '2,2,2;2,1,2;2,2,1',  '2,2,2;1,1,3;0,5,0',  '3,1,1;1,2,5;1,0,7',  '1,1,5;3,2,1;8,2,0',  '3,3,1;2,1,1;1,1,1',
'2,1,4;1,8,0;3,0,1',  '1,2,2;6,1,1;8,1,0',  '1,1,3;1,3,2;0,10,0',  '1,3,1;1,2,3;3,0,7',  '2,1,2;2,2,2;1,8,0',  '1,4,2;10,0,0;2,1,2',
'1,4,1;5,1,1;2,0,8',  '3,1,1;2,4,0;3,0,1',  '2,2,2;2,2,1;3,1,1',  '1,1,3;2,5,1;6,4,0',  '2,1,2;1,8,0;1,6,1',  '1,3,1;3,1,4;10,0,0',
'1,3,1;1,3,0;7,0,3',  '3,1,1;0,8,2;1,6,1',  '5,1,1;0,9,1;1,1,4',  '3,1,1;2,1,3;0,7,3',  '3,1,1;0,5,5;3,0,1',  '3,1,1;1,0,7;2,4,0',
'2,2,1;2,1,4;2,3,0',  '1,2,2;4,2,1;0,3,2',  '1,2,3;2,1,2;0,2,2',  '2,3,1;1,2,2;2,1,3',  '3,1,1;0,3,7;1,1,6',  '2,1,4;0,2,2;2,2,1',
'1,3,1;2,0,8;0,3,1',  '4,2,1;1,0,6;0,2,6',  '2,3,1;0,3,1;2,2,0',  '1,1,4;0,6,1;1,5,1',  '1,1,5;10,0,0;3,2,1',  '3,1,1;1,5,2;1,5,2',
'4,1,1;0,0,10;1,2,4',  '1,1,3;1,9,0;7,0,1',  '2,1,2;1,4,2;3,2,1',  '2,1,4;3,0,1;2,6,0',  '1,1,5;1,4,1;4,1,1',  '2,2,1;1,3,2;3,2,0',
'2,2,1;3,0,4;0,2,6',  '3,1,1;2,2,2;0,8,2',  '2,1,2;3,2,1;3,4,0',  '1,1,3;3,4,1;1,9,0',  '2,4,1;2,1,2;2,0,6',  '2,2,2;4,1,0;1,2,2',
'3,1,1;0,1,9;2,4,0',  '1,1,4;1,1,2;5,5,0',  '3,1,1;3,1,0;2,0,4',  '1,4,2;4,1,1;4,1,1',  '1,2,2;6,1,1;0,1,4',  '2,3,1;0,2,4;4,0,2',
'3,1,1;3,1,0;0,3,7',  '2,1,4;5,0,0;1,4,1',  '4,1,1;1,5,1;0,4,6',  '2,2,1;1,1,6;3,1,2',  '1,3,1;6,1,1;3,1,4',  '3,1,1;0,2,8;1,1,6',
'3,1,1;1,3,4;2,4,0',  '4,1,1;1,3,3;0,6,4',  '5,1,1;1,1,4;1,1,4',  '1,3,2;8,0,1;1,3,0',  '1,1,5;0,5,1;8,2,0',
'1,5,1;8,0,2;2,1,3',  '1,3,1;4,2,0;5,0,5',  '1,3,1;0,2,4;2,1,5',  '1,3,1;4,1,3;3,2,1',  '2,3,1;1,1,5;1,2,2',  '2,2,1;2,0,6;0,1,8',
'3,3,1;0,1,7;2,0,4',  '1,3,3;4,0,2;1,1,2',  '1,4,1;1,2,1;2,2,0',  '4,1,1;1,6,0;1,4,2',  '2,2,2;1,2,2;2,1,2',  '5,1,1;1,5,0;1,1,4',
'3,3,1;2,1,1;0,1,7',  '2,1,3;0,1,3;3,1,1',  '2,1,3;2,0,2;3,1,1',  '2,3,2;1,2,1;1,2,1',  '4,1,2;0,8,1;1,2,2',  '1,1,3;0,10,0;3,4,1',
'4,2,1;0,2,6;2,1,0',  '1,4,2;6,1,0;0,2,1',  '1,2,3;0,2,2;2,1,2',  '2,2,1;3,1,2;3,2,0',  '1,1,3;2,2,2;3,1,2',  '3,1,1;0,4,6;2,0,4',
'1,3,1;4,0,6;0,3,1',  '2,1,2;1,8,0;2,4,1',  '1,5,1;3,1,2;4,1,1',  '1,2,2;0,4,1;4,1,2',  '3,1,1;1,1,6;1,0,7',  '1,3,1;1,1,6;4,1,3',
'3,1,1;2,0,4;1,7,0',  '2,1,2;5,0,0;1,2,3',  '3,1,2;1,1,3;2,2,1',  '2,2,2;0,2,3;1,4,0',  '1,1,4;5,1,1;2,4,1',  '1,1,3;5,5,0;1,0,3',
'3,3,1;2,0,4;0,3,1',  '1,1,3;6,1,1;0,4,2',  '2,2,2;0,2,3;3,0,2',  '2,1,2;5,0,0;2,4,1',  '1,1,3;9,1,0;6,1,1',  '1,3,1;0,0,10;4,1,3',
'1,1,3;1,3,2;4,6,0',  '2,2,2;5,0,0;1,1,3',  '1,1,3;7,0,1;1,6,1',  '3,2,1;1,2,3;2,2,0',  '3,1,1;0,4,6;2,1,3',  '1,3,1;3,0,7;2,1,5',
'2,1,2;0,2,4;4,2,0',  '1,1,5;5,0,1;5,5,0',  '3,1,1;0,5,5;1,2,5',  '1,2,3;10,0,0;5,1,1',  '1,4,1;0,1,6;9,0,1',  '1,1,5;2,3,1;7,3,0',
'1,5,1;2,1,3;0,1,5',  '1,3,1;2,1,5;0,3,1',  '2,2,2;2,0,3;0,3,2',  '2,4,1;3,0,4;3,1,0',  '5,1,1;0,2,8;1,3,2',  '3,2,1;3,0,1;0,1,8',
'1,1,4;5,1,1;7,3,0',  '1,3,1;1,3,0;3,1,4',  '3,3,1;2,1,1;3,0,1',  '1,1,3;6,1,1;3,1,4',  '2,1,3;4,2,0;2,0,2',  '3,1,1;1,2,5;0,4,6',
'2,1,2;0,4,3;2,0,3',  '2,1,2;0,8,1;4,2,0',  '2,4,1;4,0,2;1,1,4',  '1,3,1;6,1,1;3,1,4',  '1,2,3;5,1,1;1,0,3',  '1,2,4;4,1,1;6,0,1',
'4,2,1;0,1,8;1,2,2',  '2,2,1;1,4,0;2,0,6',  '1,2,3;6,2,0;5,1,1',  '3,1,1;1,7,0;0,2,8',  '1,3,1;4,1,3;2,0,8',  '1,1,3;1,0,3;2,5,1',
'1,1,3;3,4,1;1,1,2',  '3,1,3;1,4,1;1,1,2',  '5,1,1;0,6,4;1,1,4',  '2,2,1;1,0,3;4,1,2',  '5,1,1;0,3,7;1,2,3',  '1,2,3;1,3,1;10,0,0',
'2,2,2;1,0,4;3,1,1',  '1,2,2;4,0,3;2,3,1',  '1,2,3;7,0,1;3,2,1',  '1,4,1;3,0,7;0,1,6',  '2,1,2;2,2,4;1,3,0',  '2,1,3;2,6,0;0,1,3',
'3,1,1;0,5,5;1,6,1',  '1,5,1;5,1,0;2,0,8',  '4,2,1;0,1,8;2,0,2',  '2,2,1;0,3,4;4,0,2',  '2,2,2;0,4,1;2,0,3',  '2,2,2;0,1,4;2,3,0',
'3,1,1;1,0,7;1,5,2',  '2,1,2;4,2,0;1,0,6',  '4,1,2;1,2,2;1,6,0',  '2,3,2;4,0,1;1,2,1',  '1,2,2;6,1,1;0,4,1',  '1,5,1;5,0,5;3,1,2',
'2,1,2;0,8,1;3,0,2',  '4,1,1;1,2,4;1,0,6',  '5,1,1;0,7,3;1,2,3',  '2,1,2;4,2,0;0,2,4',  '1,2,2;0,1,4;8,0,1',  '2,1,4;3,4,0;2,2,1',
'4,1,2;1,6,0;2,0,1',  '2,1,3;3,4,0;1,5,1',  '4,1,2;0,6,2;1,6,0',  '1,2,2;2,2,2;2,2,2',  '3,1,3;2,4,0;0,1,3',  '3,2,1;1,2,3;2,1,2',
'1,4,1;9,0,1;0,2,2',  '2,2,1;0,3,4;1,0,8',  '4,1,1;1,0,6;0,0,4',  '2,2,1;3,1,2;1,1,4',  '2,2,2;1,4,0;1,0,4',  '2,2,2;1,0,4;0,6,1',
'4,1,1;2,2,0;1,0,6',  '1,3,1;4,2,0;5,1,2',  '1,2,4;0,5,0;4,1,1',  '2,1,2;1,0,4;1,6,1',  '1,1,4;1,5,1;4,6,0',  '1,1,4;1,5,1;0,6,1',
'3,1,1;1,3,4;1,5,2',  '1,5,1;2,1,3;5,0,5',  '1,4,1;1,1,5;5,1,1',  '1,3,1;0,1,7;5,1,2',  '1,2,2;8,0,1;4,1,2',  '1,5,1;0,2,0;4,1,1',
'3,3,1;0,2,4;1,2,1',  '1,4,1;6,1,0;2,1,4',  '1,2,4;4,1,1;0,1,6',  '3,2,1;1,0,7;2,2,0',  '2,1,3;1,5,1;0,10,0',  '1,2,2;0,1,4;6,1,1',
'1,4,1;8,0,2;2,2,0',  '3,1,1;0,3,7;1,3,4',  '3,1,2;0,10,0;1,3,2',  '1,2,4;0,1,2;0,0,2',  '2,1,4;3,4,0;1,4,1',  '2,2,2;1,3,1;0,2,3',
'1,1,4;0,10,0;5,1,1',  '3,1,3;1,7,0;1,4,1',  '2,4,1;1,0,8;0,2,2',  '1,1,4;4,2,1;1,1,2',  '2,1,2;3,2,1;5,0,0',  '1,1,3;3,4,1;1,0,3',
'1,3,1;9,0,1;0,1,7',  '2,3,2;2,2,2;0,0,2',  '4,1,1;2,0,2;1,4,2',  '1,4,1;7,0,3;4,1,2',  '3,1,1;1,7,0;0,4,6',  '3,2,2;2,1,1;2,0,2',
'2,2,1;1,3,2;2,3,0,4',  '3,1,2;2,0,2;2,2,1',  '1,3,1;7,0,9;0,1,9',  '1,1,3;3,1,2;6,1,1',
'1,4,2;2,1,2;2,2,0',  '3,1,2;2,0,2;2,2,1',  '1,3,1;3,2,1;0,1,7',  '1,1,3;2,8,0;4,0,2',  '2,3,1;0,1,7;2,0,6',  '1,2,2;4,1,2;8,0,1',
'1,4,1;1,2,1;9,0,1',  '1,4,2;6,1,0;6,0,2',  '1,3,2;4,2,0;2,0,4',  '3,1,1;0,10,0;1,2,5',  '1,3,2;3,1,2;7,1,0',  '1,1,4;0,2,2;3,7,0',
'2,2,2;4,0,1;2,3,0',  '1,1,5;0,5,1;2,3,1',  '3,1,1;1,2,5;0,1,9',  '1,1,3;3,3,1;2,10,0,0',  '1,1,3;6,4,0;0,4,2',  '2,2,1;1,0,8;1,3,2',
'4,1,1;1,1,0;6,1,1,5',  '1,1,3;0,1,3;2,5,1',  '1,4,1;4,0,6;2,1,4',  '1,1,4;7,3,0;1,1,2',  '1,1,4;7,3,0;1,1,2',  '3,1,1;1,0,7;3,1,0',
'2,2,1;3,2,0;1,0,8',  '1,3,1;1,1,6;6,1,1',  '1,3,3;1,2,1;4,0,2',  '3,1,1;0,10,0;1,3,4',  '3,1,1;1,7,0;2,2,2',  '1,5,1;8,0,2;0,1,5',
'2,1,4;2,2,1;1,0,2',  '1,4,1;0,2,2;1,0,9',  '5,1,1;0,4,6;1,5,0',  '1,1,5;8,2,0;1,4,1',  '1,2,4;4,1,1;8,1,0',  '1,4,1;1,1,5;3,0,7',
'5,1,1;0,6,4;1,0,5',  '3,1,1;0,0,10;1,1,6',  '1,3,1;4,1,3;7,0,3',  '1,2,4;2,0,2;8,1,0',  '1,1,3;2,2,2;6,1,1',  '1,1,3;2,2,2;3,8,1,0',
'1,2,2;6,0,2;2,3,1',  '3,3,1;0,0,10;1,2,1',  '3,2,1;2,1,2;1,2,3',  '1,3,1;8,0,2;7,1,0',  '1,2,3;1,0,3;4,3,0',  '1,2,2;0,3,2;8,1,0',
'2,2,2;1,4,0;1,2,2',  '1,4,2;0,2,1;4,0,3',  '1,4,1;1,2,1;6,1,0',  '1,2,4;4,1,1;6,2,0',  '3,2,1;0,0,10;1,3,1',  '3,1,1;1,4,3;0,0,10',
'2,1,2;3,2,1;3,0,2',  '2,2,2;2,3,0;1,3,1',  '1,2,2;8,1,0;0,3,2',  '1,3,1;2,1,5;3,2,1',  '1,1,4;5,5,0;3,3,1',  '2,1,2;3,0,2;3,4,0',
'1,3,1;7,1,0;6,0,4',  '3,3,1;0,3,1;1,1,4',  '2,4,1;2,0,6;0,2,2',  '1,1,3;2,8,0;3,1,2',  '1,1,3;7,0,1;0,7,1',  '1,3,1;5,1,2;0,3,1',
'1,4,1;0,2,2;4,1,2',  '1,1,5;9,1,0;1,4,1',  '1,1,4;1,9,0;4,2,1',  '3,2,1;0,1,8;1,1,5',  '4,1,1;0,4,6;1,3,3',  '1,4,1;4,1,2;6,0,4',
'3,1,3;0,7,1;1,7,0',  '3,1,2;1,5,1;3,1,0',  '2,2,1;1,2,0;6,2,6',  '1,4,1;1,0,4;0,2,2,8,0',  '1,1,4;4,2,1;2,8,0',  '1,3,1;6,1,1;4,2,0',
'1,2,2;4,0,3;0,3,2',  '1,3,1;3,0,7;7,1,0',  '4,1,1;1,1,5;0,10,0',  '1,1,4;1,5,1;1,1,2',  '1,1,5;7,3,0;1,4,1',  '4,2,1;2,1,0;0,1,8',
'1,2,3;2,1,2;2,2,4,0',  '1,2,2;6,1,1;2,2,2',  '2,2,2;0,4,1;2,2,0',  '1,4,1;3,1,3;5,0,5',  '3,2,1;1,0,4;2,3,0,1',  '2,4,1;2,1,2;3,0,1',
'2,3,1;1,3,3;0,3,4',  '2,3,1;4,0,2;1,2,2',  '1,1,5;0,10,0;1,4,1',  '1,1,3;3,7,0;6,1,1',  '2,3,1;1,2,2;0,3,1',  '3,1,1;0,7,3;1,0,7',
'1,2,2;0,3,2;4,0,3',  '1,4,1;0,1,6;5,0,5',  '2,2,2;3,1,1;2,2,1',  '2,4,1;1,1,4;3,0,4',  '2,1,3;4,2,0;1,5,1',  '1,2,2;6,1,1;10,0,0',
'4,1,1;0,7,3;1,0,6',  '2,1,3;1,8,0;1,2,2',  '2,2,2;1,1,3;0,5,0',  '1,3,2;2,2,1;8,0,1',  '1,4,2;2,2,0;4,1,2',  '2,1,2;1,6,1;2,6,0',
'1,1,5;1,4,1;10,0,0',  '2,2,2;0,1,4;3,1,1',  '1,1,4;8,2,0;4,2,1',  '3,2,1;1,0,7;0,1,8',  '2,2,1;2,2,2;2,2,2',
'3,1,1;1,4,3;1,5,2',  '1,1,3;3,1,2;1,3,2',  '2,1,3;2,0,2;1,8,0',  '1,4,1;3,1,3;1,1,5',  '2,1,4;2,2,1;3,4,0',  '1,3,1;5,1,2;0,3,1',
'2,1,3;3,1,1;1,2,2',  '4,2,1;1,0,2;6,1,0,6',  '1,1,3;6,1,1;5,5,0',  '2,1,2;1,0,4;4,2,4,0',  '1,4,1;5,0,5;0,1,6',  '1,5,1;2,1,3;10,0,0',
'1,3,1;7,1,0;4,1,3',  '4,2,1;1,2,2;1,1,4',  '1,5,1;0,1,5;3,0,7',  '2,2,1;0,2,6;1,4,0',  '5,1,1;0,1,5;1,2,3',  '2,1,2;2,4,1;2,4,1',
'2,3,1;0,2,4;2,1,3',  '1,2,4;6,2,0;0,1,2',  '2,1,3;3,4,0;2,3,1',  '3,1,2;0,2,4;1,5,1',  '2,1,2;2,0,3;4,2,0',  '2,1,2;1,6,1;2,4,1',
'2,1,3;1,5,1;2,3,1',  '1,3,3;1,1,2;1,3,1',  '1,1,3;3,1,2;1,6,1',  '2,1,2;5,0,0;3,2,1',  '1,1,3;3,1,9,0;1,6,1',
'4,1,1;1,4,2;0,5,5',  '1,3,1;0,0,10;5,1,2',  '2,2,1;0,1,8;2,1,4',  '1,4,1;1,2,1;0,1,6',  '1,2,2;8,1,0;4,0,3',  '1,3,1;4,2,0;1,0,9',
'1,1,3;1,6,1;0,10,0',  '2,2,2;4,1,0;2,1,2',  '2,3,1;1,0,8;1,1,5',  '3,3,1;1,1,4;1,2,1',  '3,1,2;1,7,0;1,1,3',  '1,3,1;6,1,1;6,0,4',
'1,1,4;4,2,1;1,9,0',  '1,4,1;4,0,6;0,1,6',  '1,1,4;5,1,1;6,0,1',  '5,1,1;0,5,5;1,0,5',  '2,2,2;0,2,3;2,0,3',  '2,1,2;4,2,2;4,1,0',
'1,3,1;5,0,5;1,1,6',  '3,1,1;0,4,6;1,1,6',  '2,2,2;1,3,1;2,0,3',  '3,1,2;2,4,0;0,2,4',  '2,2,1;2,2,2;4,1,0',  '1,1,4;1,9,0;6,0,1',
'1,4,1;6,1,0;4,1,2',  '3,2,2;2,1,1;0,1,4',  '4,2,1;1,1,4;0,2,6',  '4,1,2;2,2,0;0,8,1',  '3,1,1;0,2,8;2,1,3',  '4,1,1;1,2,4;0,5,5',
'5,1,1;1,4,1;1,1,4',  '1,3,1;7,0,3;1,2,3',  '1,1,3;4,0,2;5,5,0',  '2,1,4;4,2,0;2,2,1',  '2,2,2;3,2,0;0,2,3',  '1,1,3;0,1,3;7,0,1',
```

```python
# --- Helper Functions ---

def _parse_quantities(q_str: str) -> Dict[str, int]:
    """Parses a quantity string like '1,2,0' into a dictionary."""
    return {item: int(q) for item, q in zip(ITEMS, q_str.split(','))}

def _format_quantities(quantities: Dict[str, int]) -> str:
    """Formats a quantity dictionary into a string like '1,2,0'."""
    return ",".join(str(quantities.get(item, 0)) for item in ITEMS)

def _create_agreement(state: State, offering_player: int, offered_quantities: Dict[str, int]) -> List[Dict[str, int]]:
    """Creates the final agreement structure based on an accepted offer."""
    shares = [{}, {}]
    shares[offering_player] = offered_quantities
    other_player = 1 - offering_player
    # The other player gets the remainder of the item pool.
    shares[other_player] = {
        item: state['pool'][item] - offered_quantities.get(item, 0) for item in ITEMS
    }
    return shares

# --- Core API Functions ---

def apply_action(state: State, action: Action) -> State:
    """Returns the new state after an action has been taken."""
    new_state = copy.deepcopy(state)
    player_id = get_current_player(new_state)

    if player_id == -1:  # Chance player sets up the game.
        pool_str, v0_str, v1_str = action.split(';')
        new_state['pool'] = _parse_quantities(pool_str)
        new_state['player_0_values'] = _parse_quantities(v0_str)
        new_state['player_1_values'] = _parse_quantities(v1_str)
        new_state['current_player'] = '0'
        return new_state

    if "agrees" in action:
        # A player agrees to the last offer, ending the game.
        last_offer = new_state['offer_history'][-1]
        new_state['agreement'] = _create_agreement(new_state, last_offer['player'], last_offer['quantities'])
        new_state['current_player'] = None  # Mark as a terminal state.

    elif "offers" in action:
        # A player makes a new offer.
        new_state['num_turns'] += 1
        quantities = _parse_quantities(action.split(' offers ')[1])
        new_offer = {
            'num_turn': new_state['num_turns'],
            'player': player_id,
            'quantities': quantities
        }
        new_state['offer_history'].append(new_offer)

        # If turn limit is reached, this offer becomes a forced, zero-reward agreement.
        if new_state['num_turns'] >= MAX_TURNS:
            new_state['current_player'] = None
            new_state['agreement'] = _create_agreement(new_state, player_id, quantities)
        else:
            new_state['current_player'] = str(1 - player_id) # Switch to other player.

    return new_state

def get_current_player(state: State) -> int:
    """Returns current player, with -1 for chance and -4 for terminal."""
    player = state.get('current_player')
    if player == 'chance':
        return -1
    if player is None:
        return -4
    return int(player)

def get_player_name(player_id: int) -> str:
    """Returns the name of the player, with 'chance' for -1, and 'terminal' for -4."""
    if player_id == -1:
        return 'chance'
    if player_id == -4:
        return 'terminal'
    return str(player_id)

def get_rewards(state: State) -> list[float]:
    """Returns the rewards per player from their last action."""
    # Rewards are only given for a voluntary agreement. A forced agreement
    # at the turn limit (MAX_TURNS) results in zero reward for both.
    if not state.get('agreement') or state['num_turns'] >= MAX_TURNS:
        return [0.0] * NUM_PLAYERS

    rewards = []
    for i in range(NUM_PLAYERS):
        player_values = state[f'player_{i}_values']
        player_share = state['agreement'][i]
        # Reward is the total value of items received by the player.
```

```python
            reward = sum(player_share.get(item, 0) * player_values.get(item, 0) for item in ITEMS)
            rewards.append(float(reward))
    return rewards

def get_legal_actions(state: State) -> list[Action]:
    """Returns all legal actions for the current player."""
    player_id = get_current_player(state)
    if player_id == -1: # Chance player
        return _CHANCE_ACTIONS
    if player_id < 0:  # Terminal state
        return []

    actions = []
    pool = state['pool']

    # Generate all possible 'offer' actions by iterating through all item combinations.
    ranges = [range(pool.get(item, 0) + 1) for item in ITEMS]
    for combo in itertools.product(*ranges):
        quantities = {item: count for item, count in zip(ITEMS, combo)}
        actions.append(f"player {player_id} offers {_format_quantities(quantities)}")

    # 'agree' is a legal move if at least one offer has been made by the opponent.
    if state['offer_history']:
        actions.append(f"player {player_id} agrees")

    return actions

def get_observations(state: State) -> list[PlayerObservation]:
    """Returns the observation for each player, containing public and private information."""
    base_obs = {
        'current_player': state['current_player'],
        'pool': state['pool'],
        'num_turns': state['num_turns'],
        'agreement': state['agreement'],
    }

    is_terminal = get_current_player(state) == -4
    observations = []

    # Determine the correct previous_offer based on game state
    terminal_previous_offer = None
    if is_terminal:
        # In a terminal state, the "previous offer" is the one that was on the table
        # before the final, accepted offer was made. This corresponds to the
        # second-to-last offer in the history.
        if len(state['offer_history']) > 1:
            terminal_previous_offer = state['offer_history'][-2]

    for i in range(NUM_PLAYERS):
        obs = base_obs.copy()
        obs['values'] = state[f'player_{i}_values']
        obs['my_player_id'] = i

        if is_terminal:
            obs['previous_offer'] = terminal_previous_offer
        else:
            # In an active game, the previous offer is the last one made by the opponent.
            opponent_id = 1 - i
            obs['previous_offer'] = next((
                offer for offer in reversed(state['offer_history']) if offer['player'] == opponent_id
            ), None)
        observations.append(obs)

    return observations

def resample_history(obs_action_history: list[tuple[PlayerObservation, Action | None]], player_id: int) -> list[Action]:
    """Stochastically samples one of many potential histories given a single player's perspective."""
    first_obs = obs_action_history[0][0]

    # Opponent's values are private and must be sampled randomly to create a possible history.
    opponent_id = 1 - player_id
    opponent_values = {item: random.randint(0, MAX_ITEM_VALUE) for item in ITEMS}

    values = [{}, {}]
    values[player_id] = first_obs['values']
    values[opponent_id] = opponent_values

    # Reconstruct the 'chance' action that started the game.
    chance_action = (
        f"{_format_quantities(first_obs['pool'])};"
        f"{_format_quantities(values[0])};"
        f"{_format_quantities(values[1])}"
    )

    # Collect all known offers (own and opponent's) from the observation history.
    known_offers = []
    seen_turns = set()
    for obs, action in obs_action_history:
        # Opponent's offers are seen in the 'previous_offer' field.
        prev_offer = obs.get('previous_offer')
        if prev_offer and prev_offer['num_turn'] not in seen_turns:
            known_offers.append(prev_offer)
```

```python
                seen_turns.add(prev_offer['num_turn'])

            # Own offers are reconstructed from the actions taken.
            if action and 'offers' in action:
                # An offer action increments the turn number for the *next* state's observation.
                # The offer itself is recorded with this new turn number.
                turn = obs['num_turns'] + 1
                if turn not in seen_turns:
                    quantities = _parse_quantities(action.split(' offers ')[1])
                    known_offers.append({'num_turn': turn, 'player': player_id, 'quantities': quantities})
                    seen_turns.add(turn)

        # Reconstruct the sequence of actions in chronological order.
        known_offers.sort(key=lambda x: x['num_turn'])
        resampled_actions = [chance_action]
        resampled_actions.extend(
            f"player {o['player']} offers {_format_quantities(o['quantities'])}"
            for o in known_offers
        )

        # Add the final action if it was not an offer (e.g., 'agrees').
        final_action = obs_action_history[-1][1]
        if final_action and 'offers' not in final_action:
            resampled_actions.append(final_action)

        return resampled_actions

def value_function(state: dict[str, Any], player_id: int) -> float:
    """Returns the value estimate for player_id in state.

    For terminal states the function returns the true return. For ongoing play
    the function should return a value estimate that reflect the winning potential
    of the player with given player_id.
    """
    # 1. Handle Terminal States
    if get_current_player(state) == -4:
        if player_id < 0 or player_id >= NUM_PLAYERS:
            # For non-players like 'terminal' (-4), return 0.0
            return 0.0
        # For active players, return the actual reward achieved.
        return get_rewards(state)[player_id]

    # --- Heuristic for Non-Terminal States ---

    # 2. Basic Information
    my_values = state.get(f'player_{player_id}_values')
    # This can happen in the initial 'chance' state before values are assigned.
    if not my_values:
        return 0.0

    opponent_id = 1 - player_id
    pool = state['pool']
    offer_history = state['offer_history']
    current_turn_player = get_current_player(state)

    # 3. Calculate Total Potential Value
    # The maximum value this player could get if they received all items.
    total_my_value = sum(pool.get(item, 0) * my_values.get(item, 0) for item in ITEMS)
    if total_my_value == 0:
        return 0.0  # If nothing in the pool is valuable, expected outcome is 0.

    # 4. Define Baseline "Fair" Expectation
    # A simple assumption that the player aims for about half the total value.
    fair_value_estimate = total_my_value / 2.0

    # 5. Core Heuristic Logic based on current negotiation status
    heuristic_value = fair_value_estimate  # Default to fair split expectation

    if not offer_history:
        # First turn, no offers yet. The best estimate is a fair split.
        heuristic_value = fair_value_estimate
    else:
        last_offer = offer_history[-1]
        if current_turn_player == player_id:
            # It's my turn to act.
            if last_offer['player'] == opponent_id:
                # Opponent made the last offer. I can agree or counter.
                # Calculate the value of their offer to me.
                their_proposed_share = last_offer['quantities']
                my_share_if_agree = {
                    item: pool.get(item, 0) - their_proposed_share.get(item, 0)
                    for item in ITEMS
                }
                value_on_table = sum(
                    my_share_if_agree.get(item, 0) * my_values.get(item, 0)
                    for item in ITEMS
                )
                # My position's value is the better of what I can get now
                # versus my general expectation from continued negotiation.
                heuristic_value = max(value_on_table, fair_value_estimate)
            else: # last_offer['player'] == player_id
```

```
        # The last offer was mine, but the opponent didn't agree.
        # My turn again means my previous offer was implicitly rejected.
        # Fall back to the baseline expectation.
        heuristic_value = fair_value_estimate
    elif current_turn_player == opponent_id:
        # It's the opponent's turn. They are considering my last offer.
        if last_offer['player'] == player_id:
            my_proposed_share = last_offer['quantities']
            # The value of the state is the value of the offer I'm hoping they accept.
            value_of_my_offer = sum(
                my_proposed_share.get(item, 0) * my_values.get(item, 0)
                for item in ITEMS
            )
            heuristic_value = value_of_my_offer

# 6. Apply Time Pressure Discount
# As turns run out, the risk of getting 0 from a forced agreement increases.
# This discounts the potential future value accordingly.
if state['num_turns'] >= MAX_TURNS:
    return 0.0 # Game is over or will be forced to 0 reward on next action.

turns_left = MAX_TURNS - state['num_turns']
# A sqrt factor makes the discount less severe in early turns.
pressure_factor = (turns_left / MAX_TURNS) ** 0.5

return float(heuristic_value * pressure_factor)
```

## J.3   BARGAINING (IMPERFECT INFORMATION, CLOSED DECK)

```python
import copy
import itertools
import random
from typing import Any, Dict, List, Tuple, Optional

# Type definitions
Action = str
State = Dict[str, Any]
PlayerObservation = Dict[str, Any]

# Game constants
NUM_PLAYERS = 2
ITEMS = sorted(['X', 'Y', 'Z'])
MAX_TURNS = 10
POOL_VALUES = range(1, 6)
ITEM_VALUES = range(0, 7)

# Create a fixed, reproducible set of possible game scenarios for the chance node.
_CHANCE_OUTCOMES = []
_chance_rng = random.Random(0)
for _ in range(20):
    pool = {item: _chance_rng.choice(POOL_VALUES) for item in ITEMS}
    p0_values = {item: _chance_rng.choice(ITEM_VALUES) for item in ITEMS}
    p1_values = {item: _chance_rng.choice(ITEM_VALUES) for item in ITEMS}

    pool_str = ",".join(f"{k}={v}" for k, v in sorted(pool.items()))
    p0_str = ",".join(f"{k}={v}" for k, v in sorted(p0_values.items()))
    p1_str = ",".join(f"{k}={v}" for k, v in sorted(p1_values.items()))
    _CHANCE_OUTCOMES.append(f"pool:{pool_str};p0_values:{p0_str};p1_values:{p1_str}")

def _parse_offer_action(action: Action) -> Tuple[int, Dict[str, int]]:
    """Parses an offer action string into player ID and quantities."""
    parts = action.split()
    player_id = int(parts[1])
    quantities = {item: int(q) for item, q in zip(ITEMS, parts[3].split(','))}
    return player_id, quantities

def _calculate_reward(bundle: Dict[str, int], values: Dict[str, int]) -> float:
    """Calculates the total value of a bundle of items for a player."""
    return sum(bundle.get(item, 0) * values.get(item, 0) for item in ITEMS)

def _reconstruct_offer_action(offer: Dict[str, Any]) -> Action:
    """Reconstructs an offer action string from an offer dictionary."""
    quantities_str = ",".join(str(offer['quantities'].get(item, 0)) for item in ITEMS)
    return f"player {offer['player']} offers {quantities_str}"

def apply_action(state: State, action: Action) -> State:
    """Returns the new state after an action has been taken."""
    new_state = copy.deepcopy(state)

    if state.get('current_player') == 'chance':
        # Initialize the game state from the chance node action.
        parts = action.split(';')
        new_state['pool'] = {p.split('=')[0]: int(p.split('=')[1]) for p in parts[0].split(':')[1].split(',')}
        new_state['player_0_values'] = {p.split('=')[0]: int(p.split('=')[1]) for p in parts[1].split(':')[1].split(',')}
        new_state['player_1_values'] = {p.split('=')[0]: int(p.split('=')[1]) for p in parts[2].split(':')[1].split(',')}
        new_state['current_player'] = 0
        return new_state

    if 'agrees' in action:
        # An agreement is reached. The game becomes terminal.
        last_offer = new_state['offer_history'][-1]
        offerer_id = last_offer['player']
        offerer_bundle = last_offer['quantities']
        accepter_bundle = {item: new_state['pool'][item] - offerer_bundle.get(item, 0) for item in ITEMS}

        agreement = [{}, {}]
        agreement[offerer_id] = offerer_bundle
        agreement[1 - offerer_id] = accepter_bundle

        new_state['agreement'] = agreement
        new_state['current_player'] = None
    elif 'offers' in action:
        # An offer is made. Increment turn count and switch player.
        player_id, quantities = _parse_offer_action(action)
        new_state['num_turns'] += 1
        offer = {'num_turn': new_state['num_turns'], 'player': player_id, 'quantities': quantities}
        new_state['offer_history'].append(offer)

        if new_state['num_turns'] >= MAX_TURNS:
            new_state['current_player'] = None  # End game if turn limit reached.
        else:
            new_state['current_player'] = 1 - player_id

    return new_state

def get_current_player(state: State) -> int:
    """Returns current player, with -1 for chance and -4 for terminal."""
    if state.get('current_player') == 'chance':
        return -1
```

```python
  if state.get('current_player') is None or state['agreement'] or state['num_turns'] >= MAX_TURNS:
      return -4
  return state['current_player']

def get_player_name(player_id: int) -> str:
  """Returns the name of the player, with 'chance' for -1, and 'terminal' for -4."""
  return {-1: 'chance', -4: 'terminal'}.get(player_id, str(player_id))

def get_rewards(state: State) -> list[float]:
  """Returns rewards. Rewards are 0 if the game ends due to the turn limit."""
  if state['num_turns'] >= MAX_TURNS or not state['agreement']:
      return [0.0] * NUM_PLAYERS

  p0_reward = _calculate_reward(state['agreement'][0], state['player_0_values'])
  p1_reward = _calculate_reward(state['agreement'][1], state['player_1_values'])

  return [float(p0_reward), float(p1_reward)]

def get_legal_actions(state: State) -> list[Action]:
  """Returns legal actions that can be taken in current state."""
  player = get_current_player(state)
  if player == -4:
      return []
  if player == -1:
      return _CHANCE_OUTCOMES

  actions = []
  if state['num_turns'] > 0:
      actions.append(f"player {player} agrees")

  # Generate all possible offer combinations based on the item pool.
  pool = state['pool']
  quantity_ranges = [range(pool.get(item, 0) + 1) for item in ITEMS]
  for quantities in itertools.product(*quantity_ranges):
      q_str = ",".join(map(str, quantities))
      actions.append(f"player {player} offers {q_str}")

  return actions

def get_observations(state: State) -> list[PlayerObservation]:
  """Returns the observation for each player."""
  observations = []
  player_at_turn = get_current_player(state)
  is_terminal = (player_at_turn == -4)

  for i in range(NUM_PLAYERS):
      previous_offer = None
      # In a terminal state with an agreement, the "previous offer" is the one before the accepted one.
      if is_terminal and state['agreement'] and len(state['offer_history']) > 1:
          previous_offer = state['offer_history'][-2]
      elif state['offer_history']:
          previous_offer = state['offer_history'][-1]

      obs = {
          'my_player_id': i,
          'pool': state['pool'],
          'values': state[f'player_{i}_values'],
          'num_turns': state['num_turns'],
          'agreement': state['agreement'],
          'previous_offer': previous_offer,
          'current_player': str(player_at_turn) if player_at_turn >= 0 else None,
      }
      observations.append(obs)
  return observations

def resample_history(obs_action_history: list[tuple[PlayerObservation, Action | None]], player_id: int, last_is_terminal: bool) ->
↪ list[Action]:
  """Stochastically sample one of many potential histories of actions for all players."""
  # 1. Reconstruct and yield the chance action.
  first_obs = obs_action_history[0][0]
  opponent_values = {'X': 3, 'Y': 3, 'Z': 4}  # Assume fixed opponent values for reproducibility.

  p_vals = [{}, {}]
  p_vals[player_id] = first_obs['values']
  p_vals[1 - player_id] = opponent_values

  pool_str = ",".join(f"{k}={v}" for k, v in sorted(first_obs['pool'].items()))
  p0_str = ",".join(f"{k}={v}" for k, v in sorted(p_vals[0].items()))
  p1_str = ",".join(f"{k}={v}" for k, v in sorted(p_vals[1].items()))
  yield f"pool:{pool_str};p0_values:{p0_str};p1_values:{p1_str}"

  # 2. Reconstruct the interleaved game actions from the player's perspective.
  last_opponent_turn_yielded = 0
  my_last_action = None
  for obs, action in obs_action_history:
      if action:
          my_last_action = action

      if obs.get('previous_offer'):
          offer = obs['previous_offer']
          # Only yield opponent offers that haven't been yielded yet to avoid duplication.
          if offer['player'] != player_id and offer['num_turn'] > last_opponent_turn_yielded:
```

```python
                yield _reconstruct_offer_action(offer)
                last_opponent_turn_yielded = offer['num_turn']

        if action:
            yield action
            if 'agrees' in action:
                return

    # 3. Deduce the final hidden actions if the game ended with an agreement not initiated by the player.
    if last_is_terminal:
        last_obs, last_action = obs_action_history[-1]
        if last_action is None and last_obs['agreement']:
            agreement = last_obs['agreement']
            _, my_last_quantities = _parse_offer_action(my_last_action)

            # Case A: Opponent agreed to my last offer. My bundle in the agreement matches my last offer.
            if my_last_quantities == agreement[player_id]:
                yield f"player {1 - player_id} agrees"
            # Case B: Opponent made a counter-offer, which I would have implicitly agreed to.
            else:
                opponent_id = 1 - player_id
                opponent_bundle = agreement[opponent_id]
                quantities_str = ",".join(str(opponent_bundle.get(item, 0)) for item in ITEMS)
                yield f"player {opponent_id} offers {quantities_str}"
                yield f"player {player_id} agrees"

from typing import Any, Dict, List

def value_function(state: dict[str, Any], player_id: int) -> float:
    """Returns the value estimate for player_id in state.

    For terminal states the function returns the true return. For ongoing play
    the function should return a value estimate that reflect the winning potential
    of the player with given player_id.
    """
    # Game constants and helper functions defined in local scope for self-containment.
    ITEMS = sorted(['X', 'Y', 'Z'])
    MAX_TURNS = 10
    NUM_PLAYERS = 2

    def _calculate_reward(bundle: Dict[str, int], values: Dict[str, int]) -> float:
        """Calculates the total value of a bundle of items for a player."""
        return sum(bundle.get(item, 0) * values.get(item, 0) for item in ITEMS)

    def _get_current_player_internal(state_dict: Dict[str, Any]) -> int:
        """Determines the current player or if the state is terminal."""
        current_player = state_dict.get('current_player')
        agreement = state_dict.get('agreement')
        num_turns = state_dict.get('num_turns', 0)

        is_terminal = (
            current_player is None or
            (agreement and isinstance(agreement, list) and len(agreement) > 0) or
            num_turns >= MAX_TURNS
        )

        if is_terminal:
            return -4  # Terminal node code
        if current_player == 'chance':
            return -1  # Chance node code
        return int(current_player)

    def _get_rewards_internal(state_dict: Dict[str, Any]) -> List[float]:
        """Calculates rewards for all players in a terminal state."""
        agreement = state_dict.get('agreement')
        num_turns = state_dict.get('num_turns', 0)

        # No reward if the game ends due to turn limit or no agreement is made.
        if num_turns >= MAX_TURNS or not agreement or (isinstance(agreement, list) and len(agreement) == 0):
            return [0.0] * NUM_PLAYERS

        p0_reward = _calculate_reward(agreement[0], state_dict['player_0_values'])
        p1_reward = _calculate_reward(agreement[1], state_dict['player_1_values'])

        return [float(p0_reward), float(p1_reward)]

    # --- Main value function logic begins ---

    current_player_code = _get_current_player_internal(state)

    # 1. Handle Terminal States: Return the exact final reward.
    if current_player_code == -4:
        if player_id < 0:  # MCTS may query the value for the terminal node itself.
            return 0.0
        return _get_rewards_internal(state)[player_id]

    # 2. Handle Non-Terminal States: Return a heuristic-based value estimate.

    my_values = state[f'player_{player_id}_values']
    pool = state['pool']
```

```python
offer_history = state.get('offer_history', [])

# Heuristic for the start of the game (no offers yet).
# A neutral assumption is that the player can achieve half of their maximum possible value.
if not offer_history:
  my_total_pool_value = _calculate_reward(pool, my_values)
  return my_total_pool_value / 2.0

last_offer = offer_history[-1]

# Case A: It's the opponent's turn. This means I made the last offer.
# The value of my last offer is a good estimate of my current potential, as it reflects my aspiration.
if current_player_code != player_id:
  my_bundle_in_my_last_offer = last_offer['quantities']
  my_value_of_my_offer = _calculate_reward(my_bundle_in_my_last_offer, my_values)
  return float(my_value_of_my_offer)

# Case B: It's my turn. The opponent made the last offer.
# My potential lies between what they offered and what I last asked for.
else:
  # Calculate the value of their offer to me. This is a concrete value I can achieve by accepting.
  offered_bundle_to_opponent = last_offer['quantities']
  implied_bundle_to_me = {
      item: pool.get(item, 0) - offered_bundle_to_opponent.get(item, 0)
      for item in ITEMS
  }
  value_of_their_offer_to_me = _calculate_reward(implied_bundle_to_me, my_values)

  # Find my last offer to gauge my own aspiration level.
  my_aspiration = -1.0
  for offer in reversed(offer_history):
    if offer['player'] == player_id:
      my_bundle_in_my_last_offer = offer['quantities']
      my_aspiration = _calculate_reward(my_bundle_in_my_last_offer, my_values)
      break

  # If I haven't made an offer yet, my aspiration defaults to the initial 50/50 baseline.
  if my_aspiration < 0:
    my_total_pool_value = _calculate_reward(pool, my_values)
    my_aspiration = my_total_pool_value / 2.0

  # The heuristic is the midpoint between their offer and my aspiration, representing a likely compromise point.
  heuristic_value = (value_of_their_offer_to_me + my_aspiration) / 2.0
  return float(heuristic_value)
```

