# OpenReview forum: "Code World Models for General Game Playing"
_ICLR.cc/2026/Conference — ICLR 2026 Poster_

### Official Review · Reviewer_2crY · 2025-10-27

**Soundness:** 1
**Presentation:** 2
**Contribution:** 3
**Rating:** 2
**Confidence:** 4

**Summary:**

This paper introduces a very interesting .

**Strengths:**

This is a thorough, well-founded work. It is well-motivated and it starts with good rigour and a clever approach to what will eventually become, in my opinion, a large open problem in agentic workflows.

**Weaknesses:**

While the motivations/contributions are strong, the paper's methodology is not there yet. It has some weaknesses that (unfortunately) do not make it, in my _personal_ opinion, ready for publication.

1. The paper's writing is inconsistent across the work. Tonal shifts occur often (e.g., L431 'you can check it in...')
2. Only one (closed-source) baseline.
    - While not strictly needed, an ablation on prompts/models would be helpful to ascertain _why_ the results are the way they are.
3. Paper structuring and presentation needs work. This is different from the paper's writing: overall the presentation / scientific methodology needs to be improved.

Here's how the paper can be improved:
1. Make the writing consistent--proof-read and ensure that it adjusts to the expected tonality of a scientific article.
    - Ensure descriptions of tables/figures are well-done. E.g., what is '# LLM Calls' in Table 2? If it is 'number of LLM calls', why is it a float? Is it an average then?
    - The related work section needs _a bit_ (not a lot) of work. Some known works in both applying Markov processes or getting LLMs to play videogames are missing. I would also suggest adding a small explanation on what they did.
    - Make sure your paper is proofread. For example, L308 introduces 'OOD' before defining what it is. The sentence in L86 does not need parentheses. This indicates a lack of proofreading.
    - Definitions are important. Gemini is not indicated as a reasoning model: since the behaviour of an LLM (next-token predictor) and an RLM (baked-in CoT) are quite distinct, referring to Gemini (an RLM) as an LLM does require some clarification.
2. More open baselines/agents will be beneficial for the robustness, soundness, and longevity of the work. These three are _musts_ for contributions to any conference, let alone ICLR.
    - Related: the code would be better put in a repository. This will avoid presentation issues like those in App. I.2
3. On scientific writing:
    1. Scientific writing follows a very specific template:
        1. Results/Experiments contain the outcomes of your evaluation of the hypothesis. These should be supported with numerical evidence. Opinions and interpretations of the results (like 5.1.1) are for the discussion.
        2. Discussion contains the discussion _of the results_, not of your paper.
        3. Conclusion (which is missing) allows you to draw a conclusion (or interpretation) of your hypothesis based on data.
    2. Significant digits must be consistent and should make sense (why is accuracy in T2 reported to five significant digits? Is such precision truly needed?).
    3. Skipping the definition of imperfect info games because they are 'tricky' _would_ make sense if it weren't for the fact that (a) the experimental work does rely on imperfect information; and (b) the appendix to which the reader is referred does not contain a formal definition.

I genuinely think this could be a very strong contribution, but needs more work than might be feasible for this submission.

Minor, and not something that influenced my review: the lack of a reproducibility statement plus (1) the fact that all the code is in an appendix; and (2) the baselines are closed-source, do not indicate good, open science practices. I would encourage the authors to rethink this and add such a statement. Again, not something that can/will impact my assessment of the paper, but always nice to have.

**Questions:**

In addition from my questions above, I'd like to know if it is possible to know what would be the behaviour of a reasoning model alone in this agentic workflow scenario. This _is_ important since a comparison between a Markov-like optimised prompt/workflow versus the effect of a (behind-an-API) reasoning model _versus_ a RLM using the Markov-optimised approaches would allow an ablation on whether it is the prompt, the workflow, or the model that provides the contributions.

---

> ### Author Response · Authors · 2025-11-20
> **Response to reviewer 2crY**
>
> Thank you for your extensive review, we provide our answers as follows:
>
> **Questions about the work**
>
> > Ablation on prompts/models:
>
> Regarding LLM ablations, we have now added synthesis results for Gemma 27B (which doesn’t use thinking) in a separate answer. Performance decreases significantly, although the approach is still able to generate the tic-tac-toe game correctly most of the time. The purpose of this work is to show that the game-playing abilities of a flagship LLM can be improved by using CWMs, vs directly using the LLM as a policy. This can be proven on a single flagship model. Even if only half of the flagship LLMs could benefit from this approach, it would nonetheless be valuable.
>
> Regarding prompt ablations, we do provide an ablation between prompting the LLM using a tree search approach and a conversational (greedy) approach. Beyond that, we haven't found any particularly insightful variations to share. We have shared the prompts we use, which have been manually tuned based on experience. Obviously, we could get better results by devoting more time and compute to systematic prompt tuning, and arbitrarily worse results with worse prompts. Regarding the prompt of the baseline, we used a comparable amount of effort to best represent the capabilities of Gemini 2.5Pro as a policy.
>
> > Some known works in both applying Markov processes or getting LLMs to play videogames are missing.
>
> We are happy to improve our related work section. Do you have any concrete suggestions beyond these?
>
> [PokéChamp: an Expert-level Minimax Language Agent](https://arxiv.org/abs/2503.04094)
> [Orak: A Foundational Benchmark for Training and Evaluating LLM Agents on Diverse Video Games](https://arxiv.org/abs/2506.03610)
> [Voyager: An Open-Ended Embodied Agent with Large Language Models](https://arxiv.org/abs/2305.16291)
>
> > Definitions are important. Gemini is not indicated as a reasoning model: since the behaviour of an LLM (next-token predictor) and an RLM (baked-in CoT) are quite distinct, referring to Gemini (an RLM) as an LLM does require some clarification.
>
> Our approach is compatible with any LLM (of course, results will vary). An RLM is a type of LLM, so referring to Gemini as an LLM is not incorrect, and by specifying that we are using Gemini 2.5Pro we are conveying even more information than if we said RLM. However, we agree that some clarifications regarding the specific settings that we use are missing. In particular, in the final version of this manuscript we will clarify that Gemini 2.5Pro is used with all tools disabled (structured outputs, code execution, function calling, grounding with Google search, etc) and with thinking enabled, with no set a thinking budget.
>
> > The code would be better put in a repository. This will avoid presentation issues like those in App. I.2
>
> Thanks for the suggestion, we commit to uploading the code in the appendix of the paper to Github upon acceptance. We can also share them anonymously beforehand if the reviewers so wish.
>
> > In addition from my questions above, I'd like to know if it is possible to know what would be the behaviour of a reasoning model alone in this agentic workflow scenario. This is important since a comparison between a Markov-like optimised prompt/workflow versus the effect of a (behind-an-API) reasoning model versus a RLM using the Markov-optimised approaches would allow an ablation on whether it is the prompt, the workflow, or the model that provides the contributions.
>
> Regarding "what would be the behavior of a reasoning model alone in this agentic workflow scenario", this is precisely what the Gemini 2.5Pro baseline is. Gemini 2.5Pro is used as a policy, and it is thinking before producing an action as an answer, i.e., it is a reasoning model. We will clarify this in the final version. Our approach uses exactly the same model, with exactly the same settings, but adds our precise workflow (CWM inference + planning) instead of directly providing an action, thus showing that the proposed workflow is what creates the difference, and not the skill of the model.

---

> > ### Author Response · Authors · 2025-11-20
> > **Response to reviewer 2crY (continued)**
> >
> > **Presentation/writing related**
> >
> > > The paper's writing is inconsistent across the work. Tonal shifts occur often (e.g., L431 'you can check it in...')
> >
> > We assume that this is in reference to "You can check its ...". We will address the reader more formally, as "The reader can check its forfeit rate". Do you have any other examples of tonal shifts (since they occur often)?
> >
> > > Ensure descriptions of tables/figures are well-done. E.g., what is '# LLM Calls' in Table 2? If it is 'number of LLM calls', why is it a float? Is it an average then?
> >
> > The notation of "# LLM calls" refers to the number of LLM calls. The results are floats because they are averaged over 5 different seeds. We will add this to the table caption.
> >
> > > Make sure your paper is proofread. For example, L308 introduces 'OOD' before defining what it is.
> >
> > OOD is defined in line 111, before it is used in L308
> >
> > > The sentence in L86 does not need parentheses. This indicates a lack of proofreading.
> >
> > L86 reads "Formally defining states in partially-observable (imperfect
> > information) games can be tricky, (...)" Parenthesis is needed, since "imperfect information" is simply an alternate way to say "partially observable", not an additional qualifier.
> >
> > > Scientific writing follows a very specific template: Results/Experiments contain the outcomes of your evaluation of the hypothesis. Opinions and interpretations of the results (like 5.1.1) are for the discussion. Conclusion (which is missing) allows you to draw a conclusion.
> >
> > We respectfully disagree on this point. It is simply not true that all proper scientific papers contain a "discussion" section and a separate "conclusion" section, and that no discussion of the results is contained within the "experiments" section. We can provide multiple examples of well-known papers not following that structure, but to give just one: "Attention is all you need" (https://arxiv.org/pdf/1706.03762) contains no discussion section, and the results are briefly discussed as part of the experiments section.
> >
> > If all reviewers were to feel strongly about this prescribed structure, we would change it.
> >
> > > Significant digits must be consistent and should make sense (why is accuracy in T2 reported to five significant digits? Is such precision truly needed?).
> >
> > We consistently provide 5 digits of precision for our accuracy measures, because we want to make apparent the distinction between games where the CWM made no mistakes (1.00000) vs made minor mistakes (e.g., 0.99932). Without many significant digits, this distinction would be lost. Reviewing all of our results, we think we can reduce this to 4 digits without losing this distinction.
> >
> > > Contrary to what is stated, a formal definition of imperfect information games is not included:
> >
> > We will include the following formal definition of imperfect information games:
> >
> > **Definition: Extensive-Form Game with Imperfect Information (Osborne & Rubinstein, 1994)**
> >
> > A finite **extensive-form game** with imperfect information is a tuple $(\mathcal{N}, \mathcal{A}, \mathcal{H}, \mathcal{Z}, \tau, \mathcal{I}, f_c, u_i)$, where:
> >
> > * $\mathcal{N} = \{ 1, 2, 3, \cdots, n\} \cup \{ c \}$ is a finite set of $n$ **players** and a special player called **chance**.
> > * $\mathcal{A}$ is a finite set of **actions**.
> > * $\mathcal{H}$ is a set of **histories** (sequences of actions), $\mathcal{Z} \subset \mathcal{H}$ the set of **terminal histories** (marking the full play of a game from start to finish). Every game starts at the empty history $h_0 = \emptyset$. At each non-terminal history $h \in \mathcal{H} - \mathcal{Z}$, let $A(h) \subseteq \mathcal{A}$ denote the set of *legal actions* available at $h$.
> > * A **player function** $\tau : \mathcal{H} - \mathcal{Z} \rightarrow \mathcal{N}$ that identifies which player is to act at every nonterminal history.
> > * For each player $i \in \mathcal{N}$, a partition $\mathcal{I}_i$ of $\{ h \in \mathcal{H} - \mathcal{Z}: \tau(h) = i \}$ with the property that for all $I \in \mathcal{I}_i$, and $h, h' \in I$: $A(h) = A(h')$ and $\tau(h) = \tau(h') = i$. $\mathcal{I}_i$ is called player $i$'s **information partition** and each $I \in \mathcal{I}_i$ is called an **information state**.
> > * A function $f_c$ that assigns a probability distribution over actions at every $h \in \mathcal{H} - \mathcal{Z}$ where $\tau(h) = c$. Here, $f_c(h) \in \Delta(\mathcal{A})$ is the **chance outcome distribution** at chance event $h$.
> > * A utility function $u : \mathcal{N} \times \mathcal{Z} \rightarrow [U^{-}, U^{+}] \subset \Re$, where $U^{-}$ and $U^{+}$ are upper and lower bounds on the utility and $u_i(z)$ for $z \in \mathcal{Z}$ is the utility to player $i$ at terminal history $z$.
> >
> > (Osborne & Rubinstein, 1994) Martin J. Osborne and Ariel Rubinstein. A Course in Game Theory. MIT Press, 1994.

---

> > > ### Comment · Reviewer_2crY · 2025-11-26
> > >
> > > Thank you for this rebuttal: I appreciate that the authors defended their points so candidly.
> > > I did overlook some things (such as the defintion of OOD; parentheses in L86) while doing my review, likely due to tiredness (so I withdraw that feedback).
> > >
> > > In terms of scientific methodology, however, I am not convinced this paper still meets the bar.
> > >
> > > 1. It is true that 'referring to an RLM as an LLM is not incorrect'; but these statements are not necessarily useful in this particular scenario given how distinct they are to traditional models. Given the issues with prompting and models, this distinction should be signposted.
> > > 2. The data and code should be released. I do appreciate the authors committing to releasing it upon publication, but they are needed for an integral peer review. While we are encouraged (not mandated) to review that material, I think that a paper reflects the author's efforts and personal work, and thus it deserves a complete analysis.
> > > 3. Using previous literature as an example does not construe a proper argument: merging certain structures muddles interpretation of the experimental results: the argument of asking all reviewers to change it also does not work here since we all asked for distinct things (which is part of the review process; not everyone covers the same things)
> > > 4. The works suggested are good, but I think a thorough literature review are the responsibility of the authors to frame the work properly. Since the paper still does not contain the updates, I assume that this point still stands. This also applies for writing: I appreciate the authors requesting more examples, but I would argue it is their responsibility to ensure it is proofread.
> > > 5. Prompting--as the authors state--'obviously' could likely yield different results. Not necessarily better, since, as the authors note, hand-designed these prompts. Variance on other prompts should be indicated nonetheless in the paper, not in the rebuttal forum.
> > > 6. Proving something on a flagship model is not extensible to other models: pretraining, code, post-training, all could and do influence results. Otherwise the results are for this specific model (and the full paper should be framed as such; not as a proof-of-concept).

---

### Official Review · Reviewer_jLRx · 2025-10-31

**Soundness:** 4
**Presentation:** 3
**Contribution:** 3
**Rating:** 8
**Confidence:** 3

**Summary:**

This paper proposes Code World Models (CWMs), using an LLM to translate natural-language rules plus a few example trajectories into executable Python code that implements a game’s world model (state transitions, legal moves, termination), along with synthesized heuristic value functions and inference functions for imperfect-information games. Experiments cover 5 perfect information games and 5 imperfect information games, and showing promising results while using Gemini 2.5 Pro as the LLM.

**Strengths:**

1. "data to code" fashion creates a verifiable simulation of games, and both perfect and imperfect information games can be applied in this fashion.
2. Two ways of synthesizing inference functions for imperfect information games to avoid the exponential cost is a valuable contribution. Both hidden history inference and hidden state inference are straightforward yet effective.

**Weaknesses:**

1. Synthesize quality relies on generated test cases over a limited amount of trajectories. When LLM failed to parse game rules, it might not build the code world model effectively.
2. It seems like CWM performs worse than Random in the game of Gin rummy.

**Questions:**

1. In closed-deck learning, can you quantify how the learned state-space size correlates with performance? This corresponds to your hypothesis at Line 455 - 457.
2. Have you tried iterative online refinement (updating the CWM during play), and if so, does it reduce reliance on high-quality initial trajectories? (You note it’s possible but skipped for efficiency.)

---

> ### Author Response · Authors · 2025-11-20
> **Response to reviewer jLRx**
>
> Thank you for your review and insightful questions. Let us address them in the following.
>
> > In closed-deck learning, can you quantify how the learned state-space size correlates with performance? This corresponds to your hypothesis at Line 455 - 457.
>
> For reference, here is the mentioned hypothesis: "We hypothesize that the non-intuitive improvement of CWM-ISMCTS-Closed at Hand of war w.r.t. the open deck setting could be due to the freedom to synthesize simpler state spaces when playing closed deck."
>
> Our phrasing was ambiguous and will be corrected. By "simpler state spaces" we didn't necessarily mean state spaces with a smaller size, but state spaces in which the game description was simpler, enabling more efficient planning (for instance, by having a smaller number of chance nodes between observation nodes). In principle, in the case of deterministic, fully observable games, the size of the state space shouldn't affect the MCTS process, and we wouldn't expect to see a correlation between state-space size and performance. However, in the case of stochastic, partially observable games, we can obtain "simpler" state spaces that simplify planning. To clarify with an example: Consider a partially observable game with a 50-dimensional hidden space that involves throwing 5 dice in each turn. The outcome of the first die impacts the first 10 dimensions of the state space, the outcome of the second die the next 10 dimensions, and so on. However, the observations and game dynamics are mostly contained in the first 10 dimensions of the state space. When learning closed-deck, we could learn a "simpler" state space that only contains the first 10 dimensions, and correspondingly simpler dynamics that only use the first die. Planning in this simplified game with chance nodes of cardinality $6$ instead of $6^5$ will be more performant (since the cardinality of the action space heavily impacts a fixed budget MCTS), even if the dynamics are slightly inaccurate. In contrast, when learning open-deck, we would have to respect the original 50-dimensional state space and the presence of 5 chance nodes, with combined cardinality $6^5$. This will result in a more correct CWM, but one for which MCTS will be much slower. For a fixed planning budget, the first one might perform better.
>
> > Have you tried iterative online refinement (updating the CWM during play), and if so, does it reduce reliance on high-quality initial trajectories? (You note it’s possible but skipped for efficiency.)
>
> We have conducted some preliminary testing of iterative online refinement. What we have observed in these tests is that for some games it is possible to completely eliminate the need for initial trajectories. I.e., it is possible to start with a CWM generated exclusively from the game's textual description and then refine the CWM online as we play. Also, for very complex games, an online approach can provide more informative trajectories for refinement, since they are on-policy. If seeking maximum sampling efficiency, this might be the best approach. However, in practice, interleaving planning and synthesis might make the overall CWM acquisition process slower without a significant performance gain.
>
> We will include these additional clarifications and explanations in the final version of the manuscript.

---

> > ### Comment · Reviewer_jLRx · 2025-11-26
> >
> > The authors address my concerns. Thus, I will maintain my rating.

---

### Official Review · Reviewer_4ELJ · 2025-11-01

**Soundness:** 2
**Presentation:** 3
**Contribution:** 2
**Rating:** 8
**Confidence:** 3

**Summary:**

The authors extend the Code World Model (CWM) framework by considering two-player games, performing value function code synthesis to improve player performance, introducing the concept of "inference as code" to enable state estimation in imperfect information games, and providing a learning algorithm (based on code-based autoencoders) to enable learning in the novel closed deck (strict partial observability) setting. Their results shows the superiority of this approach with respect to LLMs as policies on multiple perfect and imperfect information games, including newly created ones.

**Strengths:**

- Well written paper.
- Very good related works section
- Good amount of environments/games tested

**Weaknesses:**

- Less ablation studies performed
- Very few baselines
- Only Gemini 2.5 tested. Other LLMs would bring the variance that is needed to be demonstrated

**Questions:**

There are many who have shown "Code as policies", working in different settings. Why is it better than just creating actions through LLMs?

---

> ### Author Response · Authors · 2025-11-20
> **Response to reviewer 4ELJ**
>
> Thank you for your review and giving us the opportunity to expand on some of our design decisions. We address your comments in the following.
>
> > Less ablation studies performed
>
> We want to highlight the ablations that we have performed, some of which are presented only in the appendices and might not be immediately obvious: 1) For CWM refinement: Tree-search vs conversation (greedy), 2) For planning at play time: MCTS vs PPO; 3) For the initial MCTS value estimation: Inferring a value function vs standard leaf value initialization. While more ablations would no doubt be useful, these are the ones that we deemed most relevant to the novel elements introduced in this paper. Perhaps the most glaring missing ablation is one that varies the LLM used, which we address in the next response.
>
> > Very few baselines. Only Gemini 2.5 tested. Other LLMs would bring the variance that is needed to be demonstrated.
>
> We have now added synthesis results for Gemma 27B (which doesn’t use thinking) in a separate answer. Performance decreases significantly, although the approach is still able to generate the tic-tac-toe game correctly most of the time. The proposed method is LLM-agnostic and could be run on any modern LLM. Therefore, we agree that it would be very interesting to see how its performance varies across many different LLMs (we ourselves are curious), but there are some reasons for this work to not include this:
>
> - In this work we pit against each other two different ways to use an LLM to play games: A more direct one that uses the LLM as a policy and a more elaborate one that creates a CWM describing the game and plans with it. We think that we were able to convincingly show that _at least one of the flagship LLMs_ could benefit from the more elaborate process and was able to generate code describing games, including OOD games that could not possibly have been in its training data. We would expect that, regardless of absolute deviations in performance across different LLMs, the benefit of our more elaborate method persists. Although it would have been ideal to be able to verify this fact as part of this work, it would have involved other tradeoffs, as we explain next.
> - Using LLMs is expensive and somewhat slow, so our work was constrained. We chose to spend our time and budget showing the benefits of our approach on a larger number of games (and for different variations of the method), providing solid results for one LLM, rather than spreading ourselves too thin over many LLMs.
>
> > There are many who have shown "Code as policies", working in different settings. Why is it better than just creating actions through LLMs?
>
> While code as a policy is indeed a viable way to approach game playing using LLMs, we chose CWMs to approach general game playing because (a) fewer game play data is needed and (b) more powerful generalizations are possible.
>
> To see (a), consider a simple, fully observable, deterministic novel game for which you have a textual description. Writing a CWM is, in principle, as simple as translating that description into code, which an LLM could potentially do in a single attempt, with zero data. Even if the LLM doesn't succeed at first, each observed transition in that game can be used as a verifiable reward to refine the CWM. Once the CWM is correct, we can use powerful planning methods on it without needing any additional data. In contrast, knowing the rules of the game doesn't directly tell us much about how the code of a good policy should look like. Instead, we will need multiple complete rollouts -not individual transitions- to even start to get a sense for whether a given code policy is actually good and refine it. Regarding (b), it is clear that small modifications of the game (e.g., negate the reward function, or change how a chess piece moves) result in small modifications of the CWM, but potentially big changes in the policy. These modifications can again be zero-shot in a CWM-based approach, but could require almost complete retraining from scratch in a code-as-policy approach.

---

### Official Review · Reviewer_Liqi · 2025-11-01

**Soundness:** 3
**Presentation:** 2
**Contribution:** 3
**Rating:** 6
**Confidence:** 2

**Summary:**

This paper proposes a new method for gameplay agents, to use LLMs to generate a code world model, which serves as a verifiable simulation engine for the planning algorithms. The experiments evaluate the gameplay performance on 10 different games, outperforming or rivaling Gemini 2.5 Pro.

**Strengths:**

1. To use the LLM to generate the code world model is a novel idea to represent a specific game in a verifiable manner.

2. The work is solid. I also read the appendix of the paper. It provides all necessary details and examples.

**Weaknesses:**

1. The experiments are made on 10 distinct games, generalizing 4 to the other 6 games. It is hard for readers to assess the OOD generalizability of the method.

2. One concern is what kind of games (e.g. poker-like games) can benefit from the proposed method. Can the method work for any types of games? The complexity to generate a code world for card games is somewhat low, for example, the code cannot be very long, so what about more complex games?

**Questions:**

Please see above.

---

> ### Author Response · Authors · 2025-11-20
> **Response to reviewer Liqi**
>
> Thank you for your review, let us answer your questions and provide some clarifications:
>
> > The experiments are made on 10 distinct games, generalizing 4 to the other 6 games. It is hard for readers to assess the OOD generalizability of the method.
>
> The in-distribution vs out-of-distribution (OOD) question is an important one when dealing with LLMs that have been exposed to vast amounts of data during training. We want to first clarify that our approach does not rely on learning from some games and applying that knowledge to some other novel games. Instead, our proposed method is applied _independently_ to each of the 10 different games, and does not involve any training or finetuning of Gemini 2.5Pro. Some of the games in which we evaluate our method are "in distribution", meaning that they are well-known games that Gemini 2.5Pro could have conceivably encountered during training, including its textual descriptions, code, or even gameplay trajectories. One could argue that Gemini 2.5Pro could be creating the Code World Model for those games simply by recalling its training data, which would be less interesting. To make sure that this is not what is happening, we created _novel games_ whose code is not available online for LLM training. Some of these games are modifications of other existing, well-known games (e.g., generalized chess), but others have been entirely created from scratch by us (e.g, Quadranto, see Section H.9 in page 45). Quadranto is an example of a completely OOD game for which our approach still works. The results on these games support the generalizability of the method. Of course, as we move towards more obscure/complicated game styles, we might need to increase the number of offline trajectories (currently set at 5) that are necessary to fully infer the mechanics of the game, or, at some point, struggle to learn the game altogether.
>
> > One concern is what kind of games (e.g. poker-like games) can benefit from the proposed method. Can the method work for any types of games? The complexity to generate a code world for card games is somewhat low, for example, the code cannot be very long, so what about more complex games?
>
> Although in principle the proposed method is general, it does indeed incorporate some biases that makes some games easier to learn, e.g.: (a) games whose rules and code are more similar to those in the LLM training set; (b) games that can be succinctly described in Python (possibly with the use use of well-known libraries); (c) games whose stochasticity can be compactly expressed as discrete "chance node actions" (e.g., card draws, dice rolls, etc). None of these biases preclude general game learning, but as you correctly pointed out, they make learning some games easier than others. For the games in this work, we were able to use as little as 5 offline trajectories to induce the CWM. As games become more complex, inferring the CWM would require more offline trajectories, an LLM that is able to satisfactorily handle longer contexts, and the planning algorithm would require longer runtimes for the agent to play well. These types of limitations are inherent to the complexity of the problem and are likely to be shared by any general playing method.
>
> We will modify the paper to mention the presence of these biases and which factors can be used to assess how well this method will work on other novel games beyond the ones presented in this paper.

---

### Author Response · Authors · 2025-11-20
**New synthesis results using Gemma 27B (open-weights LLM)**

We have repeated our synthesis method for perfect information games and closed-deck imperfect information games using the open-weights model Gemma 27B. The results are as follows:

### Table 1: Perfect information games
**Caption:** Refinement via conversation using Gemma 27B, averages over 5 seeds.

| Game | OOD | Transition Acc. (Train) | Transition Acc. (Test) | # LLM Calls |
| :--- | :---: | :---: | :---: | :---: |
| Backgammon | ✗ | 0.17462 | 0.16742 | 500.0 |
| Connect Four | ✗ | 0.07826 | 0.10005 | 500.0 |
| Tic-tac-toe | ✗ | 0.98049 | 0.98441 | 193.0 |
| Gen. tic-tac-toe | ✓ | 0.94833 | 0.90307 | 500.0 |
| Gen. chess | ✓ | 0.49281 | 0.51603 | 500.0 |

### Table 2: Imperfect information games
**Caption:** Hidden history inference, closed deck using Gemma 27B, averages over 5 seeds.

| Game | OOD | Inference Acc. (Train) | Inference Acc. (Test) | # LLM Calls |
| :--- | :---: | :---: | :---: | :---: |
| Bargaining | ✗ | 0.09412 | 0.14296 | 500.0 |
| Leduc poker | ✗ | 0.13333 | 0.11799 | 500.0 |
| Gin rummy | ✗ | 0.00797 | 0.01393 | 500.0 |
| Quadranto | ✓ | 0.09412 | 0.10343 | 500.0 |
| Hand of war | ✓ | 0.07812 | 0.07949 | 500.0 |

### Discussion

Gemma 27B is an open-weights language model with no thinking capabilities. As expected, Gemma 27B is significantly less performant at extracting Code World Models (CWM). For tic-tac-toe, out of 5 seeds, we obtain the perfect CWM in 4 seeds. For the remaining games, the correct CWM is never extracted, although a non-trivial amount of progress is achieved for Generalized tic-tac-toe. We didn’t produce arena results, since with such poor CWMs it is obvious that the resulting agents won’t perform well.

As expected, our method works best with the most performant coding models. Although more performant models can also produce better actions when used directly as a policy, results can be further improved by using our proposed approach.

---

### Public Comment · ~Bevis_Jame1 · 2026-03-03

This paper presents a fascinating approach to leveraging Large Language Models (LLMs) in game playing by creating explicit code world models. The shift from using LLMs as direct policy generators to employing them for formal code generation is a significant advancement. The advantages of verifiability, strategic depth, and generalization are compelling, especially in the context of imperfect information games. It's exciting to see how this method not only enhances performance but also provides a robust framework for understanding game mechanics. https://stickmanhook3.io/

---

### Meta-Review · Area_Chair_eJwq · 2026-01-05

**Summary:**

This is a very interesting work to use the LLM to translate natural language rules and game trajectories into a formal, executable world model represented as Python code. Basically, the idea is novel and interesting. The validation is thorough. I do think this should be an accept with strong contribution.
One suggestion: this paper is also using LLM to generate python code as value function to guide (self-play) the MCTS search for game playing, it is highly related and suggested to be discussed in the final version:
Strategist: Self-improvement of LLM Decision Making via Bi-Level Tree Search.
The Thirteenth International Conference on Learning Representations (ICLR'25)
https://openreview.net/pdf?id=gfI9v7AbFg

**Reviewer Concerns:**

The reviewer who gives negative feedbacks are majorly concerning about the writing and experiments, such as "Make the writing consistent--proof-read" and "More open baselines/agents will be beneficial for the robustness, soundness".
In the rebuttal, the majority concerns, in my opinion, are addressed. Some concerns are not because of the reviewer is not included in the discussion, such as:
"Some known works in both applying Markov processes or getting LLMs to play videogames are missing.

==>We are happy to improve our related work section. Do you have any concrete suggestions beyond these?"

**Reviewer Scores:**

The majority give positve scores, only one give score 2, but some concerns are addressed, some are not because the reviewer is not included in the discussion.

---

### Decision · Program_Chairs · 2026-01-26

Accept (Poster)